# STAT3 promotes IFNγ/TNFα-induced muscle wasting in an NF-κB-dependent and IL-6-independent manner

Jennifer F Ma[1], Brenda J Sanchez[1], Derek T Hall[1], Anne-Marie K Tremblay[1], Sergio Di Marco[1] & Imed-Eddine Gallouzi[1,2,*] iD

## Abstract

Cachexia is a debilitating syndrome characterized by involuntary muscle wasting that is triggered at the late stage of many cancers. While the multifactorial nature of this syndrome and the implication of cytokines such as IL-6, IFNγ, and TNFα is well established, we still do not know how various effector pathways collaborate together to trigger muscle atrophy. Here, we show that IFNγ/TNFα promotes the phosphorylation of STAT3 on Y705 residue in the cytoplasm of muscle fibers by activating JAK kinases. Unexpectedly, this effect occurs both in vitro and in vivo independently of IL-6, which is considered as one of the main triggers of STAT3-mediated muscle wasting. pY-STAT3 forms a complex with NF-κB that is rapidly imported to the nucleus where it is recruited to the promoter of the iNos gene to activate the iNOS/NO pathway, a well-known downstream effector of IFNγ/TNFα-induced muscle loss. Together, these findings show that STAT3 and NF-κB respond to the same upstream signal and cooperate to promote the expression of pro-cachectic genes, the identification of which could provide effective targets to combat this deadly syndrome.

**Keywords** inflammation; iNOS; muscle wasting; NF-κB; STAT3
**Subject Categories** Cancer; Musculoskeletal System

## Introduction

Cancer-related cachexia is a debilitating syndrome characterized by the progressive loss of body weight that is triggered in part by an involuntary loss of skeletal muscle mass, referred to as muscle wasting (Dodson et al, 2011; Ma et al, 2012; Argiles et al, 2016). Cachexia is associated with anorexia, fatigue, skeletal muscle weakness, and an overall reduced quality of life (Tisdale, 2009; Argiles et al, 2014a,b). Unlike muscle atrophy caused by starvation or physical inactivity, muscle wasting in cachectic patients cannot be reversed by nutritional supplementation (Fearon, 2011; Argiles et al, 2013). Furthermore, the development of cachexia is a strong predictor of poor treatment outcome and mortality for individuals afflicted with cancer, and is estimated to be a direct cause of 20% of patient death (Muscaritoli et al, 2015). Cancer cachexia is largely considered to be an end-of-life condition. Despite this, there is no standard treatment option available to prevent or treat cachexia. This highlights the importance of delineating the pro-cachectic mechanism of action in order to identify targets to combat this deadly syndrome.

The primary catabolic mediators of cachexia include proinflammatory cytokines such as tumor necrosis factor alpha (TNFα), interferon gamma (IFNγ), and interleukin-6 (IL-6) (Hall et al, 2011; Argiles et al, 2013; Cohen et al, 2015). Numerous studies have been dedicated to develop pharmacological inhibitors to interfere with their pro-cachectic effects both in animals and humans. However, the targeted inhibition of these cytokines, specifically in human, was not efficacious. For example, the use of anti-TNFα inhibitors not only had limited effect on the progression of this condition but also has been associated with unwanted side effects (Wiedenmann et al, 2008; Jatoi et al, 2010). Additionally, recent clinical trials using a monoclonal anti-IL-6 antibody on patients with lung cancer-induced muscle wasting showed a reversal of anorexia, fatigue, and anemia, but did not prevent the loss of lean body mass (Bayliss et al, 2011). While no anti-IFNγ therapy has been tested in humans so far, some studies using antibodies against this cytokine have shown some successes in interfering with cancer-induced muscle loss in mice (Langstein et al, 1991; Matthys et al, 1991). Inhibition of cytokines may also impinge on their primary roles in the immune response. Indeed, it has been reported that the prolonged suppression of cytokines such as IL-6 or TNFα can result in increased risk of infection and delayed wound healing (McFarland-Mancini et al, 2010; Jones et al, 2011). The limited success of these mono-therapeutic approaches underscores the multifactorial nature of cachexia and highlights the possibility that its pathology could result from the combined activation of common downstream effector pathways.

1 Department of Biochemistry, Rosalind and Morris Goodman Cancer Centre, McGill University, Montreal, QC, Canada
2 Life Sciences Division, College of Science and Engineering, Hamad Bin Khalifa University (HBKU), Education City, Doha, Qatar
*Corresponding author. Tel: +1 514 398 4537, +97444546416; Fax: +1 51 398 7384; E-mails: imed.gallouzi@mcgill.ca; igallouzi@hbku.edu.qa

During the last few decades, several downstream signaling pathways and their effectors have been identified and linked to cytokine-induced muscle loss (Cohen *et al*, 2015). It has been shown that TNFα in collaboration with IFNγ mediates its pro-cachectic effects through the activation of the NF-κB (Nuclear transcription Factor kappa B) pathway, which promotes the expression of several target genes (Dahlman *et al*, 2010; Grivennikov & Karin, 2010; Hall *et al*, 2011; Fearon *et al*, 2012; Ma *et al*, 2012; Argiles *et al*, 2016). Others and we have demonstrated that the inducible nitric oxide (NO) synthase (iNOS) enzyme is one of the main effectors of the TNFα/NF-κB-induced muscle wasting (Buck & Chojkier, 1996; Di Marco *et al*, 2005, 2012). The upregulation of the iNOS/ NO pathway correlates with a dramatic decrease in general translation and the loss of promyogenic factor such as MyoD and Myogenin (Hall *et al*, 2011; Ma *et al*, 2012). More recently, several studies have indicated that IL-6 triggers muscle atrophy by activating the STAT3 (Signal transducer and activator of transcription 3) pathway (Bonetto *et al*, 2011, 2012; Sala & Sacco, 2016; Zimmers *et al*, 2016). IL-6 activates signal transduction by binding to the IL-6 receptor alpha-chain and the common receptor subunit gp130. The gp130-associated Janus kinases (JAKs) become activated and mediate the phosphorylation of STAT3 protein on its key tyrosine 705 (Y705) residues. This JAK-mediated phosphorylation allows STAT3 to dimerize and bind to DNA to promote the transcription of several pro-cachectic genes (Zhang *et al*, 2009; Bonetto *et al*, 2011, 2012; Sala & Sacco, 2016; Zimmers *et al*, 2016). Interestingly, IL-6 is not the only activator of STAT3 during muscle wasting. Treating *Il6*$^{-/-}$ mice with lipopolysaccharides (LPS), a component of Gram-negative bacteria, also leads to phosphorylation of STAT3 (Bonetto *et al*, 2012). While these observations raise the possibility that STAT3 can trigger muscle wasting independently of IL-6, a STAT3-mediated and IL-6-independent molecular mechanism remain elusive.

Several studies have shown that NF-κB and STAT3 can collaborate together to mediate cell response to various extracellular challenges. For example, unphosphorylated STAT3 was shown to associate with the p65 subunit of NF-κB to promote the transcription of the serum amyloid A (SSA) gene, the expression of which is associated with serious complications of inflammatory diseases, such as rheumatoid (RA) arthritis and juvenile inflammatory arthritis (Gillmore *et al*, 2001; Hagihara *et al*, 2005). It has also been shown that TNFα activates STAT3 in various cell systems in an NF-κB-dependent manner (Guo *et al*, 1998; Lee *et al*, 2013; Snyder *et al*, 2014). TNFα stimulates metastatic pathways in breast cancer cells by triggering the formation of the STAT3-NF-κB complex, which in turn, upregulates the transcription of the actin-bundling protein fascin (Snyder *et al*, 2014). Therefore, it is possible that the failure of the anti-TNFα and -IL-6 therapies to prevent muscle wasting is due to the fact that both cytokines redundantly activate a common downstream effector such as STAT3. In this work, we show that STAT3 is required for the IFNγ/TNFα-induced muscle wasting and that these effects depend on the collaboration between STAT3 and NF-κB pathways. Interestingly, while this STAT3-NF-κB-mediated effect occurs independently of IL-6, NF-κB is required for the translocation of STAT3 to the nucleus as well as for the activation of the iNOS/NO pathway, one of the main effectors of IFNγ/TNFα-induced muscle wasting.

## Results

### TNFα and IFNγ activate STAT3 in muscle fibers in an IL-6-independent manner

As a first step in assessing the role of the STAT3 pathway in IFNγ and TNFα-induced muscle wasting, we treated fully differentiated muscle fibers with these two cytokines over various period of times and assessed muscle wasting as well as STAT3 activation (Fig 1). While as expected (Guttridge *et al*, 2000) IFNγ or TNFα separately did not promote muscle wasting of C2C12 myotubes (Fig 1C), both cytokines together did trigger a dramatic loss of muscle fibers within 72 h of treatment (Fig 1A and B; Di Marco *et al*, 2012, 2005). The observed IFNγ/TNFα-induced muscle wasting was not due to cell death by apoptosis, since these cytokines failed to trigger the cleavage of caspase-3, a well-known marker of caspase-mediated apoptosis (Beauchamp *et al*, 2010; von Roretz *et al*, 2013), in C2C12 myotubes even after 72 h of treatment (Appendix Fig S1). Next, we assessed the activation of STAT3 in muscle fibers exposed to both cytokines as described above. We followed the phosphorylation status of the tyrosine (Y) 705 and serine (S) 727 residues, two well-characterized phosphorylation sites of STAT3 (Grivennikov & Karin, 2010). While IFNγ and TNFα did not affect the level of pS-STAT3 when compared to untreated muscle fibers, a 30-min treatment with these two cytokines was sufficient to significantly enhance the levels of pY-STAT3 in C2C12 myotubes (Fig 1D and E). Similar results were obtained when we assessed pY-STAT3 levels in primary muscle fibers treated with IFNγ/TNFα (Appendix Fig S2). Together, these observations raise the possibility that the phosphorylation of STAT3 on its Y705 residue is an integral part of the downstream signaling pathway used by IFNγ/TNFα to trigger muscle wasting.

Work from several groups has established pY-STAT3 as a one of the key mediators in the IL-6-induced muscle atrophy (Bonetto *et al*, 2011, 2012; Zimmers *et al*, 2016). Since IFNγ/TNFα have been demonstrated to induce IL-6 expression and secretion in various cell lines (Sanceau *et al*, 1991; Alvarez *et al*, 2002), we wanted to verify whether their ability to activate STAT3 in muscle fibers depends on IL-6 expression and secretion. As a first step, we analyzed the levels of IL-6 secreted by muscle fibers treated with IFNγ/TNFα as described above. Using ELISA assay, we detected a dramatic increase in IL-6 protein levels 24 h and 48 h post-treatment with amounts ranging from ~2 to ~14 ng/ml (Fig 2A). Next, we tested whether these levels of IL-6 are sufficient to induce STAT3 phosphorylation in muscle fibers. These fibers were treated with various concentration of murine recombinant IL-6 (rIL-6, ranging from 1 to 20 ng/ml) that reflected the approximate concentrations detected by ELISA during the IFNγ/TNFα treatment. Western blot analysis using the anti-pY-STAT3 antibody indicated no change in the levels of pY-STAT3 with any of these concentrations of recombinant IL-6 (Fig 2B and C). However, pY-STAT3 was detected only when C2C12 muscle fibers or macrophages (used as positive control) were treated with 100 ng/ml of IL-6 (Appendix Fig S3). Therefore, the failure of low doses of rIL-6 (20 ng/ml or lower) to phosphorylate STAT3 in myotubes might be due to instability or inactivity of our rIL-6. To further explore the effect of IL-6 on STAT3 phosphorylation, we assessed pY-STAT3 levels in the muscle of IL-6 knockout (KO) and wild-type mice, which were intramuscularly injected with IFNγ/ TNFα or saline for 5 days as described (Di Marco *et al*, 2012).

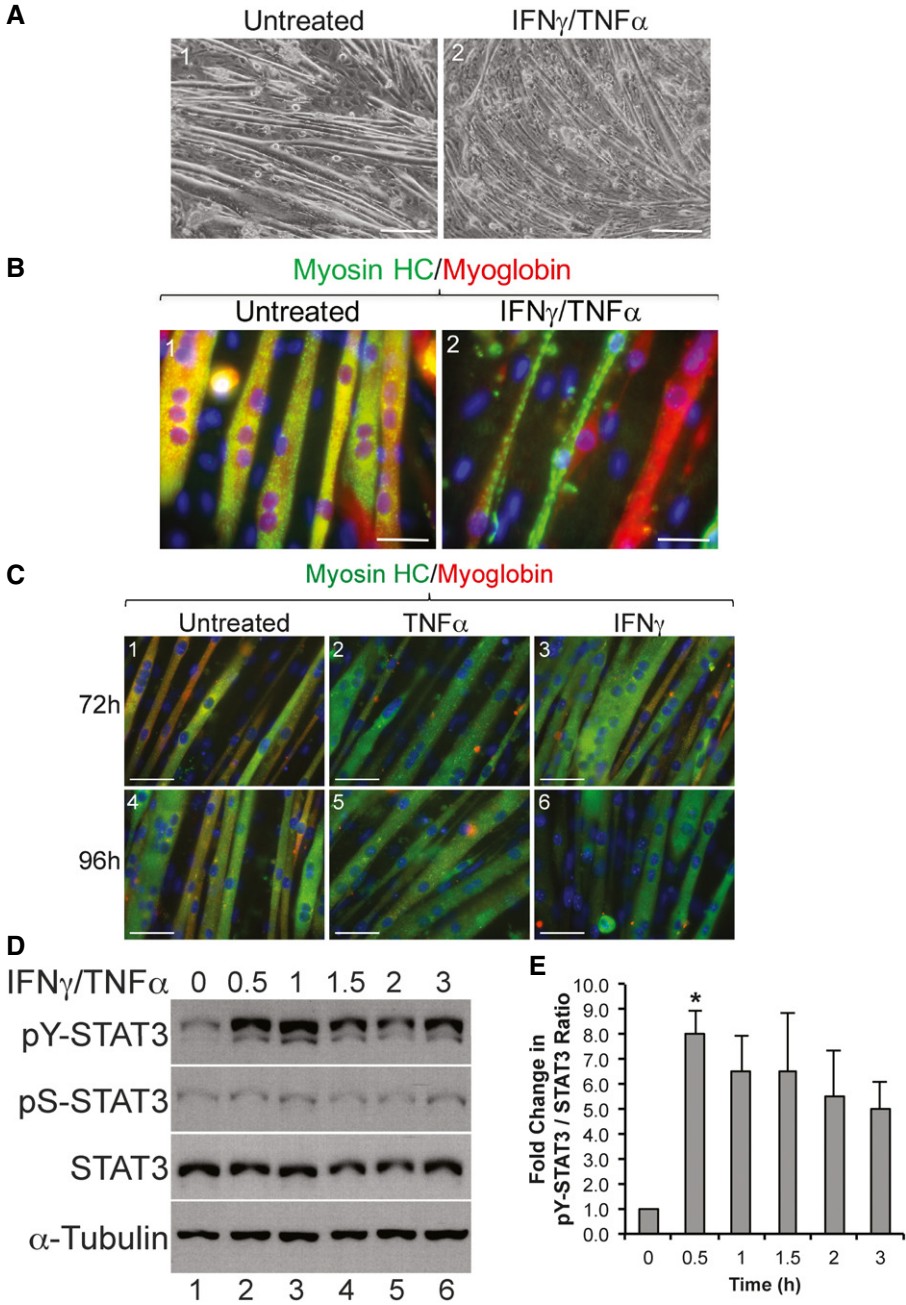

**Figure 1. STAT3 is phosphorylated on residue Tyr705 during IFNγ/TNFα-induced muscle wasting.**

C2C12 cells were grown to confluency then induced for differentiation for 4 days to form fully differentiated myotubes. These myotubes were exposed or not to IFNγ/TNFα for 72 or 96 h.

A   Phase contrast images of cultured C2C12 myotubes treated with or without IFNγ/TNFα for 72 h. Scale bar = 200 μm. Images shown are representative of *n* = 4 independent experiments.

B   Immunofluorescence (IF) images of C2C12 myotubes either untreated or treated with IFNγ/TNFα for 72 h were stained with anti-myosin heavy-chain (MyHC; green) and anti-myoglobin (red) antibodies. Scale bar = 50 μm. Images shown are representative of *n* = 4 independent experiments.

C   IF images of C2C12 myotubes treated for 72 h (panels 1–3) or 96 h (panels 4–6) with either TNFα (panels 2 and 5) or IFNγ (panels 3 and 6). Untreated (panels 1 and 4) and treated myotubes were stained with myosin heavy chain (green) and myoglobin (red) as marker for differentiated muscle cells. Images shown are representatives of *n* = 2 independent experiments. Scale bar = 50 μm.

D   Total extracts of C2C12 myotubes treated with or without IFNγ/TNFα were used for Western blot analysis with antibodies against pY-STAT3, pS-STAT3, total STAT3, and α-tubulin. The blot shown is a representative of *n* = 4 independent experiments.

E   Densitometric quantification of pY-STAT3 signal relative to total STAT3 signal in panel (D). Data represented as mean ± SEM (*n* = 4) with *P-value = 0.0135 by one-way ANOVA with Dunnett's *post hoc* test.

Source data are available online for this figure.

Interestingly, we observed a significant induction of pY-STAT3 in the gastrocnemius muscle of both wild-type and IL-6 KO animals (Fig 2D and E). However, unlike in wild-type mice, IFNγ/TNFα failed to trigger the wasting of skeletal muscle of IL-6 KO mice as evidenced by the weight and cross-sectional analyses of the tibialis anterior of these animals (Fig 2F and H). Therefore, our data demonstrate that in muscle fibers IFNγ/TNFα activate STAT3 by triggering the phosphorylation of its Y705 residue in an IL-6-independent manner.

### IFNγ/TNFα use the JAK signaling pathway to phosphorylate STAT3 and induce muscle wasting

Next, we assessed the possibility that STAT3 activation could play an important part in IFNγ/TNFα-induced muscle wasting. Since it is well established that knocking down STAT3 expression prevents myogenesis *in vitro* (Sun *et al*, 2007; Wang *et al*, 2008), to assess this possibility, we used the STAT3 inhibitor S3I-201 on the C2C12 myotubes. As has been shown in other cell lines (Pang *et al*, 2010), we observed that this inhibitor was effective in interfering with IL-6-mediated phosphorylation of STAT3 on its Y705 residue in muscle cells (Fig 3A and B). Phase contrast images as well as immunofluorescence experiments on muscle fibers exposed or not to IFNγ/TNFα for 72 h showed that S3I-201 significantly reduced the muscle fiber loss that is normally observed with these two cytokines (Fig 3C–E). To identify the signaling pathway behind the IFNγ/TNFα-mediated activation of STAT3, we assessed the involvement of the Janus kinases (JAK's), an upstream non-receptor tyrosine kinases that, in response to IL-6 and other stimuli, phosphorylate STAT3 on its Y705 residue (Grivennikov & Karin, 2010). We observed that the Jak kinase inhibitor 1 (Pedranzini *et al*, 2006) was sufficient to prevent both STAT3 phosphorylation (Fig 3F) and the IFNγ/TNFα-induced muscle atrophy (Fig 3G and H). The widths of the muscle fibers simultaneously treated with a Jak inhibitor and IFNγ/TNFα were also significantly higher than the IFNγ/TNFα-treated DMSO control fibers (Fig 3I). These results indicate that IFNγ/TNFα trigger muscle wasting by activating the JAK signaling pathway, which in turn phosphorylates STAT3 on its Y705 residue.

### pY-STAT3 promotes iNOS expression during IFNγ/TNFα-mediated muscle wasting

Previously, iNOS/NO pathway was shown as an important mediator of IFNγ/TNFα-induced muscle wasting (Di Marco *et al*, 2005, 2012). In addition, work from various groups has suggested that STAT3 modulates iNOS expression in non-muscle cells (Yu *et al*, 2002b; Ziesche *et al*, 2007; Park *et al*, 2010). Therefore, we assessed the involvement of STAT3 in IFNγ/TNFα-mediated iNOS expression in C2C12 muscle fibers. First, as expected (Di Marco *et al*, 2005), when used separately, TNFα but not IFNγ was able to slightly induce iNOS expression in C2C12 myotubes when used for 48 h or more. However, when both cytokines were used together for the same period of time, they synergistically induced a high expression level of iNOS in these muscle fibers (Appendix Fig S4). Next, we observed that S3I-201 completely inhibited iNOS expression and NO production in muscle fibers treated with IFNγ/TNFα (Fig 4A–C and Appendix Fig S5). Moreover, an shRNA that reduced STAT3 expression by ~40% also decreased iNOS protein levels by > 35% in

muscle fibers treated with IFNγ/TNFα (Fig 4D). To further explore the importance of STAT3 phosphorylation, we assessed the effect of two STAT3 mutants on iNOS expression. We observed that while expressing a constitutively active isoform of STAT3 (STAT3-C) (Bromberg & Darnell, 1999) in C2C12 cells failed, on its own, to promote iNOS expression (Fig 4E, lanes 5 and 6), this isoform dramatically enhanced the level of IFNγ/TNFα-induced iNOS protein. However, the expression of a Tyr705 to Phe (Y705F)-STAT3 mutant (Wen & Darnell, 1997) dramatically decreased iNOS expression under these conditions (Fig 4E). Additionally, using iNOS KO mice, we observed that although these animals are protected against IFNγ/TNFα-induced loss of muscle mass (Fig 4F–I), a high level of pY-STAT3 was detected in their gastrocnemius muscle (Fig 4G). These data show that while activating STAT3 alone is not sufficient to trigger iNOS expression, STAT3 collaborates with other signaling pathways to promote IFNγ/TNFα-induced iNOS and muscle atrophy. Moreover, these observations also indicate that iNOS is a key downstream effector of the IFNγ/TNFα-induced muscle wasting *in vivo*.

It is known that in addition to the NF-κB binding sites, the murine iNOS promoter has binding sites for STATs, IRFs, and AP-1 (Aktan, 2004). The binding of STAT3 to the murine iNOS promoter has been demonstrated to have, depending on the cell type and the stimuli, both a positive and a negative effect on the transcription of the *iNos* gene (de la Iglesia *et al*, 2008; Yu *et al*, 2009; Puram *et al*, 2012). We, therefore, performed ChIP coupled to qPCR experiments to determine whether, similar to what has been shown before in other cell systems (Yu *et al*, 2002a), the effects of STAT3 on iNOS expression during IFNγ/TNFα-induced muscle wasting occur through a direct binding of STAT3 to the iNOS promoter (Fig 5A). We observed a ~2.5-fold increase in the recruitment of STAT3 and p65 subunit of NF-κB to the iNOS promoter region in IFNγ/TNFα-treated muscle fibers (Fig 5B–E). The recruitment of STAT3 and p65 to the iNOS promoter was similar to those of RNA polymerase II under these conditions (Fig 5F). These results indicate that in response to IFNγ/TNFα pY-STAT3 and p65 binds to the iNOS promoter to induce its transcription in muscle fibers undergoing wasting.

### IFNγ/TNFα induces the translocation of pY-STAT3 to the nucleus in an NF-κB-dependent manner

The ability of STAT3 to activate transcription relies on its localization from the cytoplasm to the nucleus upon stimulation. Therefore, we assessed the localization of pY-STAT3 in muscle fibers treated with or without IFNγ/TNFα for the indicated periods of time. We observed, by immunofluorescence (IF), a substantial increase in the localization of pY-STAT3 to the nucleus as early as 0.5 h post-treatment with IFNγ/TNFα (Fig 6A). Nuclear and cytoplasmic fractionation experiments followed by Western blot analysis showed that the significant increase in the level of pY-STAT3 in the nucleus induced by IFNγ/TNFα persists for the first few hours of the treatment (Fig 6B and C). The increase in pY-STAT3 levels in the nucleus is not due to contamination between the fractions, since as expected hnRNPA1 is only detected in the nucleus, α-tubulin is only detected in the cytoplasm, while β-actin, as previously described (Bettinger *et al*, 2004), was detected in both fractions (Fig 6B).

The ability of STAT3 and NF-κB to cooperatively regulate the transcription of common genes is known to occur through direct

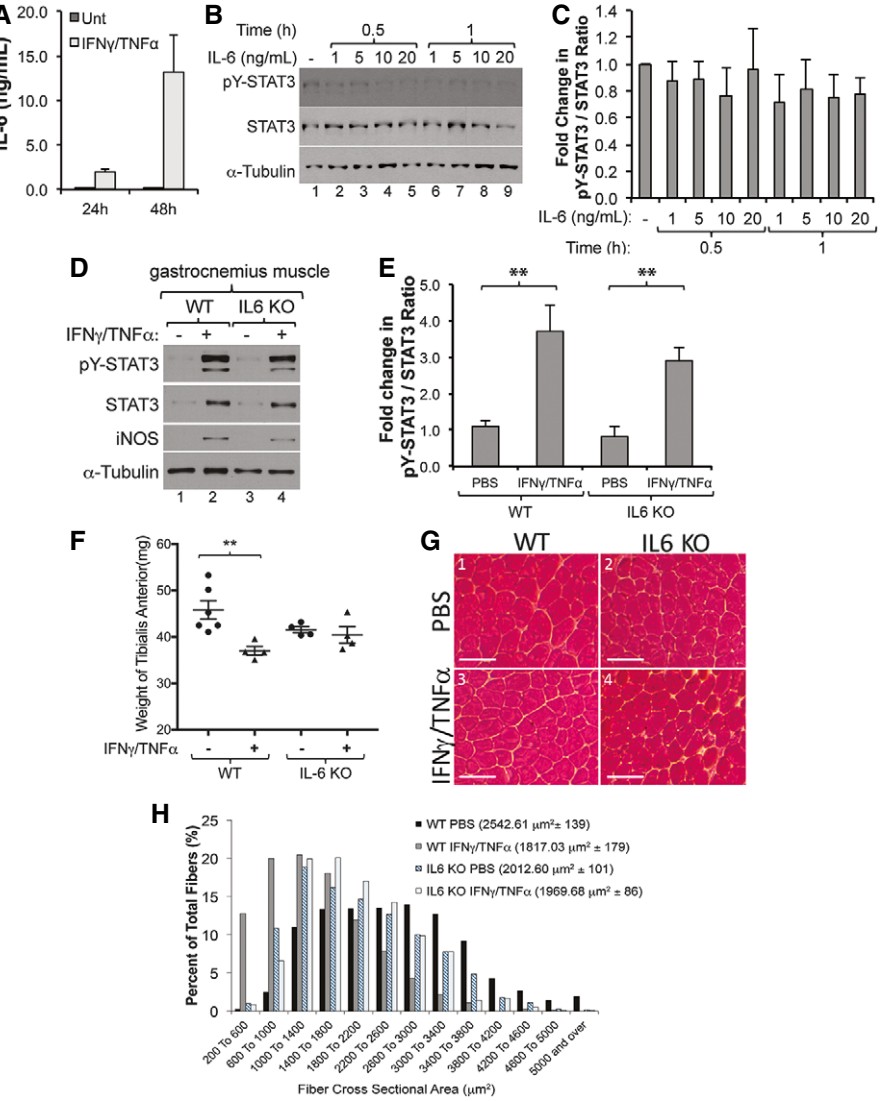

**Figure 2. IFNγ/TNFα induces STAT3 phosphorylation in an IL-6-independent manner.**

A Supernatant was collected from C2C12 myotubes treated with or without IFNγ/TNFα and analyzed by ELISA to determine the concentration of IL-6. Data represented as mean ± SEM (n = 2).

B Total cell extracts of C2C12 myotubes treated with or without recombinant murine IL-6 (rIL-6) were used for Western blot analysis with antibodies against pY-STAT3, total STAT3, and α-tubulin. The blot shown is a representative of n = 3.

C Densitometric quantification of pY-STAT3 signal relative to total STAT3 signal from Western blots in panel (B). Data represented as mean ± SEM (n = 3) with P-value = N.S. by two-way ANOVA with a Tukey post hoc test.

D Wild-type (WT) and IL-6 KO mice were intramuscularly injected with IFNγ/TNFα for five consecutive days and sacrificed on the sixth day. Gastrocnemius muscle was homogenized and used for Western blot analysis with antibodies against pY-STAT3, total STAT3, iNOS, and α-tubulin. The blot shown is a representative of n = 3 mice.

E Densitometric quantification of pY-STAT3 signal relative to total STAT3 signal from Western blots in panel (D). Data represented as mean ± SEM (n = 3) with **P-value = 0.0033 (WT) and **P-value = 0.0091 (IL-6 KO) by two-way ANOVA with a Tukey post hoc test.

F The tibialis anterior (TA) muscle weight significantly decreased in WT animals but not in IL-6 KO animals. Data represented as mean ± SEM for n = 6 (WT PBS), 4 (WT IFNγ/TNFα), 4 (IL-6 KO PBS), and 4 (IL-6 KO IFNγ/TNFα) mice with **P-value = 0.0081 by two-way ANOVA with a Tukey post hoc test.

G Image of a representative section of the TA muscle from wild-type and IL-6 KO mice stained with hematoxylin and eosin. Scale bar = 100 μm.

H The cross-sectional areas (CSA) of TA muscle from panel (G) are represented as a frequency histogram from n = 2 mice. Nine hundred fibers were analyzed for each animal. The mean CSA ± SD is indicated in the legend of the histogram.

Source data are available online for this figure.

interaction with each other (Grivennikov & Karin, 2010). Using immunoprecipitation of STAT3 or the p65 subunit of NF-κB, we detected an interaction between STAT3 and NF-κB in both the nuclear and cytoplasmic fractions of cytokine-treated or cytokine-untreated muscle fibers (Fig 7A). This observation raises the possibility that the formation of the STAT3-NF-κB complex occurs

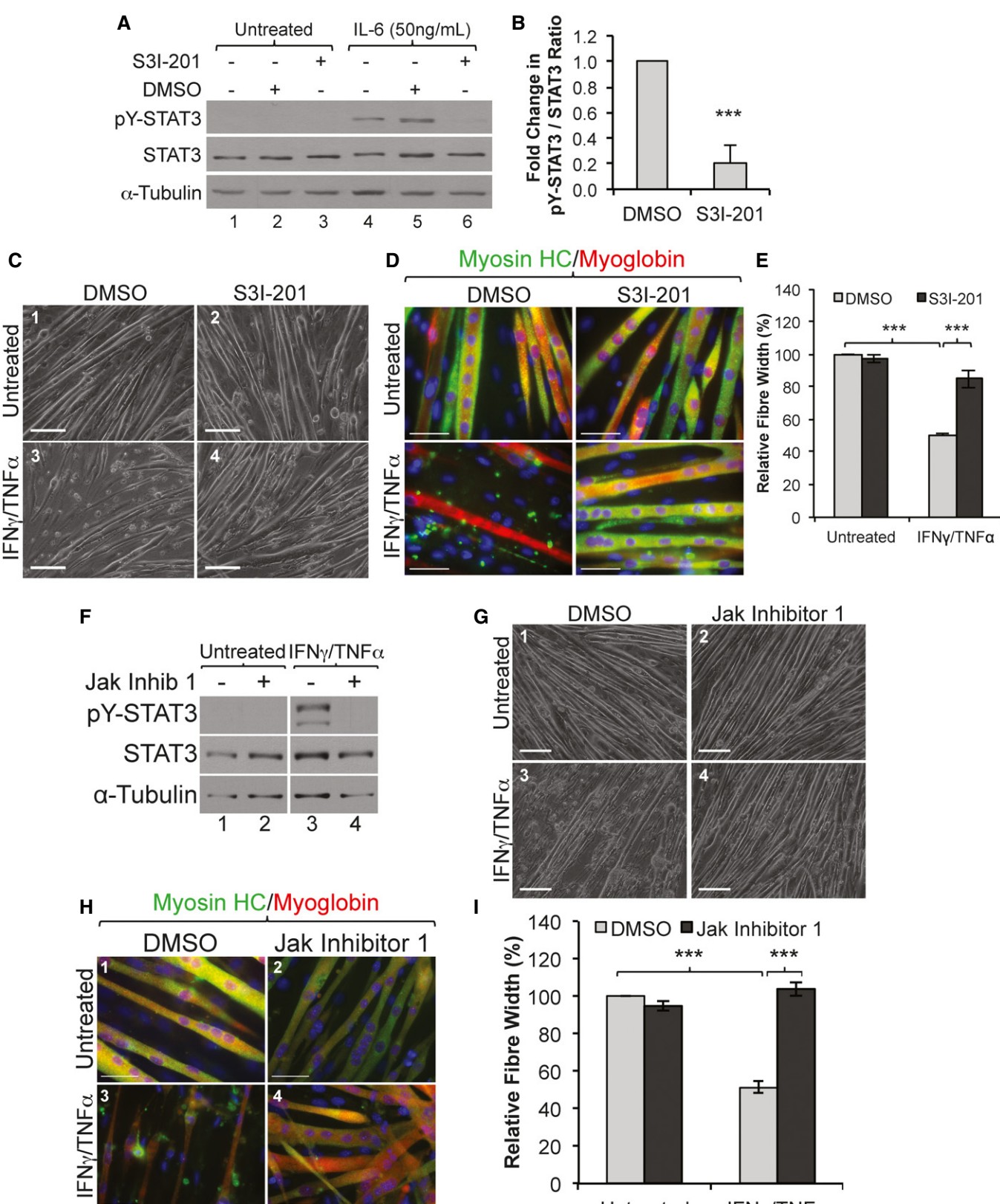

**Figure 3.**

**Figure 3.  Inhibition of the Jak/STAT3 pathway prevents IFNγ/TNFα-induced muscle wasting.**

A    Total cell extracts from proliferating C2C12 treated with STAT3 inhibitor S3I-201 then with recombinant murine IL-6 for 30 min were used for Western blot analysis with antibodies against pY-STAT3, total STAT3, and α-tubulin. The blot shown is a representative of $n$ = 3 experiments.

B    Densitometric quantification of pY-STAT3 signal relative to total STAT3 signal of Western blots in panel (A). Data represented as mean ± SEM ($n$ = 3) with a ***$P$-value = 0.00439 by two-tailed, unpaired Student's $t$-test.

C, D    Phase contrast (C) and IF images (D) of C2C12 myotubes treated with or without IFNγ/TNFα and with STAT3 inhibitor S3I-201 or DMSO as a control. IF of C2C12 myotubes were stained with antibodies against endogenous myosin heavy chain (green), myoglobin (red), and counter stained with DAPI for nuclei (blue). Scale bar is 200 μm and 50 μm for images in (C) and (D), respectively. Images shown are representatives of $n$ = 3 experiments.

E    Quantification of average myotube width from panel (D). The average myotube widths are presented as percentage change ± SEM ($n$ = 3) relative to the untreated, with ***$P$-value = $5.6 \times 10^{-5}$ (DMSO) and ***$P$-value = $7.56 \times 10^{-4}$ (S3I-201) by two-way ANOVA with Tukey post-test.

F    Total extracts from C2C12 myotubes pre-treated with a Jak inhibitor then with IFNγ/TNFα for 30 min was used for Western blotting analysis with antibodies against pY-STAT3, total STAT3, and α-tubulin. The blot shown is a representative of $n$ = 3 experiments.

G, H    Phase contrast images (G) and IF (H) of C2C12 myotubes treated with or without a Jak inhibitor or DMSO as a control and with or without IFNγ/TNFα for 72 h. IF images (H) of cultured C2C12 myotubes stained for myosin heavy chain (green), myoglobin (red), and counter stained with DAPI for nuclei (blue). Scale bar is 200 μm and 50 μm for images in (G) and (H), respectively. Images shown are representative of $n$ = 3 experiments.

I    Quantification of C2C12 myotube widths from panel (H) were determined as described in panel (D). Data represented as average ± SEM ($n$ = 3), with ***$P$-value = $1.8 \times 10^{-5}$ (DMSO) and ***$P$-value = $2.3 \times 10^{-5}$ (Jak inhibitor) by two-way ANOVA with Tukey post-test.

Source data are available online for this figure.

in the cytoplasm before translocation to the nucleus. It is well established that the retention of NF-κB in the cytoplasm occurs through its interaction with IκBα (Grivennikov & Karin, 2010). To assess whether the IFNγ/TNFα-induced nuclear translocation of STAT3 depends on the activation of the NF-κB pathway, we used muscle fibers stably over-expressing mutant NF-κB inhibitor protein IκBα (super repressor cells; SR) which cannot be phosphorylated or subsequently degraded. As a result, NF-κB cannot translocate to the nucleus and remains inactive in the cytoplasm (Guttridge *et al*, 2000; Di Marco *et al*, 2005). We observed that unlike in control fibers, IFNγ/TNFα treatment of SR muscle fibers resulted in a decrease in the levels of both pY-STAT3 and p65 subunit of NF-κB in the nucleus (Fig 7B). Additionally, Western blot analysis and immunofluorescence experiments showed that the inhibition of the NF-κB pathway with Bay 11-7082 prevented the IFNγ/TNFα-induced

nuclear translocation of pY-STAT3 as well as iNOS expression in muscle fibers undergoing wasting (Fig 7C and D). Taken together, these data demonstrate that NF-κB is required for the nuclear translocation of pY-STAT3 and for STAT3-mediated activation iNOS expression during IFNγ/TNFα-induced muscle wasting.

## Discussion

The failure of anti-TNFα or anti-IL-6 therapies to prevent muscle wasting in humans is a clear indication that the onset of this deadly syndrome is resistant to mono-therapeutic intervention (Wiedenmann *et al*, 2008; Jatoi *et al*, 2010; Bayliss *et al*, 2011). However, it is not known if this failure is due to activation of unique molecular pathways sufficient to induce atrophy, or if pro-cachectic

**Figure 4.  STAT3 promotes the expression of iNOS in IFNγ/TNFα-treated myotubes.**

A    C2C12 myotubes treated with or without IFNγ/TNFα and with STAT3 inhibitor S3I-201 or DMSO as a control. Total cell extracts from these muscle fibers were used for Western blot analysis with antibodies against iNOS, STAT3, and α-tubulin as a loading control. The blot shown is representative of $n$ = 3 experiments.

B    NO levels were measured in supernatant from the myotubes described in panel (A). Data represented as mean ± SEM ($n$ = 3) with **$P$-value = 0.0013 and ***$P$-value = 0.0006 by two-tailed, unpaired Student's $t$-test.

C    Total mRNA from C2C12 myotubes treated with IFNγ/TNFα and with the STAT3 inhibitor S3I-201 or DMSO as a control was used for RT–qPCR analysis of iNOS (*Nos2*) and *Rpl32* mRNA. Data represented as mean ± SEM ($n$ = 3) with ***$P$-value = 0.0007 by two-tailed, unpaired Student's $t$-test.

D    C2C12 cells were transfected with a plasmid expressing shRNA against STAT3 or a scramble control. (Left) Total cell extract was used for Western blot analysis with antibodies against pY-STAT3, total STAT3, iNOS, and α-tubulin. Densitometric quantification of STAT3 (middle panel) and iNOS (right panel) signals relative to α-tubulin of Western blots from left panel. Data represented as mean ± SD ($n$ = 2) with *$P$-value = 0.0123 by two-tailed, unpaired Student's $t$-test.

E    C2C12 cells were transfected with an empty vector or a STAT3 constitutively active (STAT3-C) mutant or a Tyr705 to Phe (Y705F) mutant-expressing plasmid and followed by treatment with TNFα alone, IFNγ alone, or both for 24 h. Total cell extracts were used for Western blot analysis using antibodies against pY-STAT3, total STAT3, iNOS, and α-tubulin. The blot shown is a representative of $n$ = 2 experiments.

F–I    Wild-type (WT) and iNOS KO mice were intramuscularly injected with IFNγ/TNFα for five consecutive days and sacrificed on the sixth day.

F    Wild-type (WT) and iNOS KO mice were intramuscularly injected with IFNγ/TNFα for five consecutive days and sacrificed on the sixth day. The gastrocnemius muscle weight significantly decreased in WT animals but not in iNOS KO animals treated with IFNγ/TNFα. Data represented as mean ± SEM ($n$ = 7 (WT, PBS), 4 (WT, IT), 6 (iNOS KO, PBS), and 5 (iNOS KO, IT) mice) with *$P$-value = 0.0210 by two-tailed, unpaired Student's $t$-test.

G    Wild-type (WT) and iNOS KO mice were intramuscularly injected with IFNγ/TNFα for five consecutive days and sacrificed on the sixth day. Gastrocnemius muscle was homogenized and used for Western blot analysis with antibodies against pY-STAT3, total STAT3, iNOS, and α-tubulin. The blot shown is a representative of $n$ = 3 (WT PBS), 2 (WT IFNγ/TNFα), 3 (iNOS KO PBS), and 2 (iNOS KO IFNγ/TNFα) mice.

H    Wild-type (WT) and iNOS KO mice were intramuscularly injected with IFNγ/TNFα for five consecutive days and sacrificed on the sixth day. Image of representative sections of the gastrocnemius muscle from wild-type and iNOS KO mice stained with hematoxylin and eosin. Scale bar = 100 μm. The image shown is a representative of gastrocnemius muscles from each group ($n$ = 2).

I    Wild-type (WT) and iNOS KO mice were intramuscularly injected with IFNγ/TNFα for five consecutive days and sacrificed on the sixth day. The CSA of gastrocnemius muscles from panel (G) is represented as a frequency histogram. One thousand fibers were analyzed for each group ($n$ = 2). The mean CSA ± SD is indicated in the legend of the histogram.

Source data are available online for this figure.

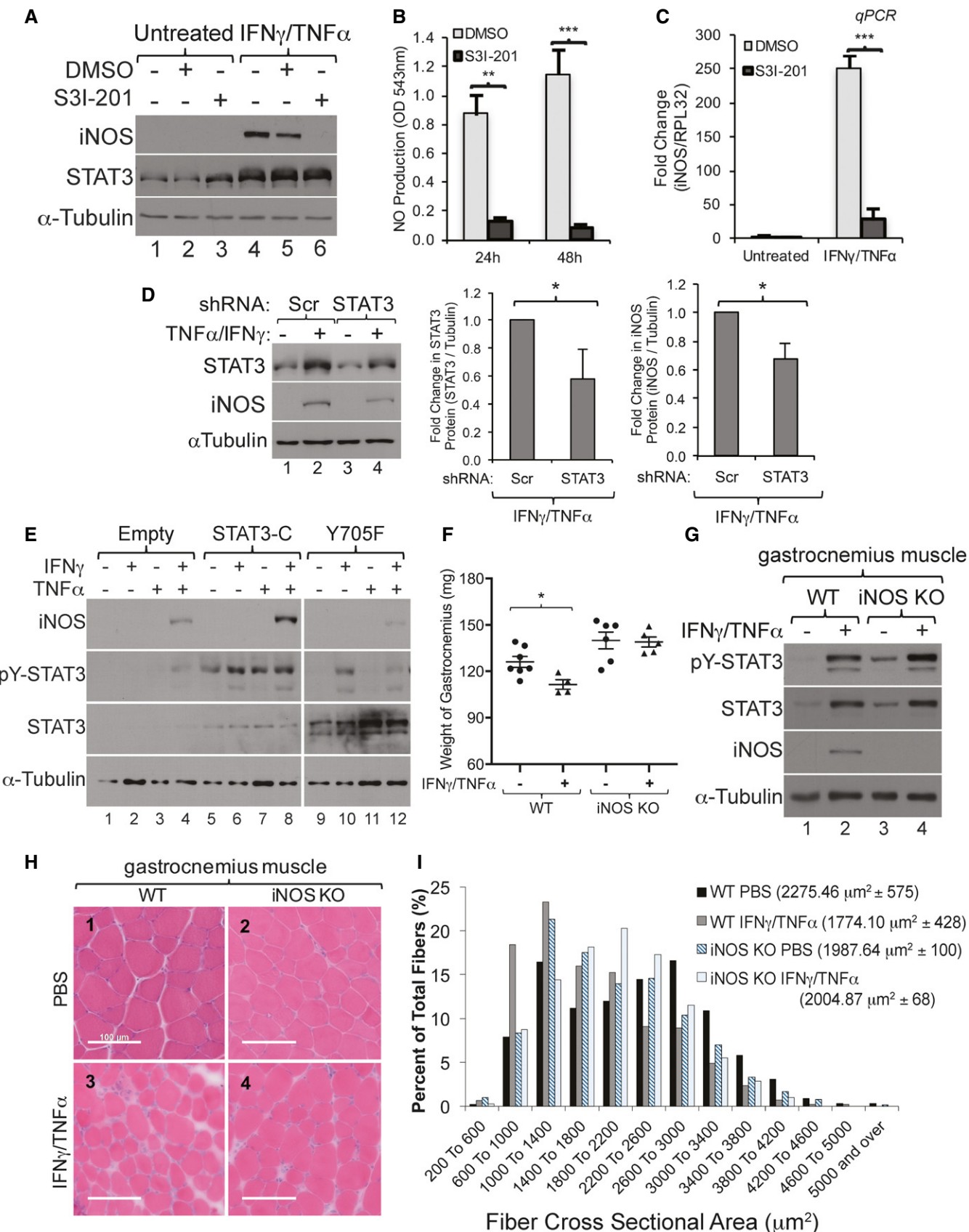

Figure 4.

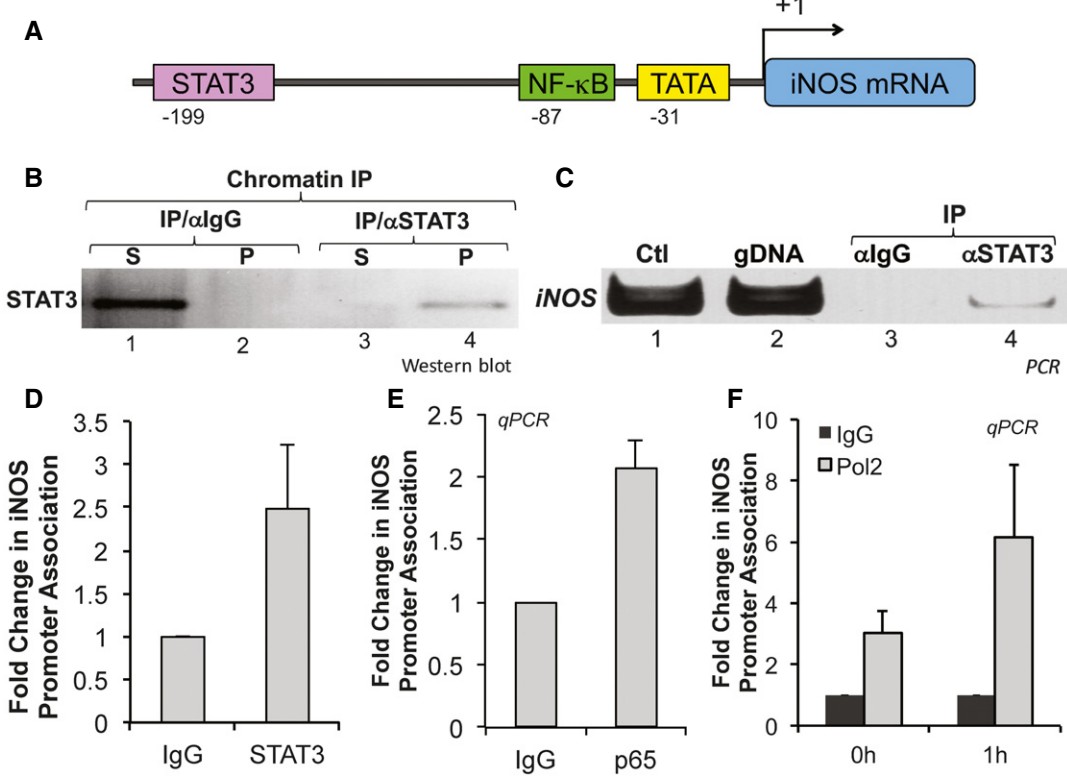

**Figure 5.  STAT3 and NF-κB bind to the iNOS promoter during cytokine-induced muscle wasting.**

A   Diagram of the iNOS promoter depicting known binding sites for STAT3 and NF-κB.

B   Chromatin prepared from C2C12 myotubes treated with or without IFNγ/TNFα was used for chromatin immunoprecipitation (ChIP). Supernatants (S) and pellets (P) of the ChIP were used for Western blot with antibodies against STAT3α or p65 or IgG control.

C   DNA from ChIP assay with anti-STAT3 (STAT3α) and IgG antibodies was analyzed by PCR using primers for the iNOS promoter. Ctl: plasmid containing iNOS promoter. gDNA: genomic DNA.

D   DNA from ChIP assay with anti-STAT3 (STAT3α) and IgG antibodies was analyzed by qPCR using primers for the iNOS promoter. Data represented as mean ± SD of $n$ = 2 experiments.

E   DNA from ChIP assay with anti-p65 (p65α) and IgG was analyzed by qPCR using primers for the iNOS promoter indicating that the iNOS promoter is associated with STAT3α. Data represented as mean ± SD ($n$ = 2).

F   DNA from ChIP assay with anti-RNA polymerase II was analyzed using primers for the iNOS promoter, indicating that RNA polymerase II is recruited to the iNOS promoter in response to IFNγ/TNFα. Data represented as mean ± SD ($n$ = 2).

Source data are available online for this figure.

signals collaborate to activate common downstream effector targets. In this study, we show that STAT3 collaborates with the NF-κB pathway to promote IFNγ/TNFα-induced muscle wasting. While the implication of STAT3 in muscle atrophy has been mainly associated with IL-6 function (Bonetto *et al*, 2011, 2012; Zimmers *et al*, 2016), our data demonstrate that to trigger muscle fiber loss, IFNγ/TNFα rapidly phosphorylates STAT3 on its Y705 residue. IFNγ/TNFα mediates this effect by activating the JAK kinases independently of IL-6. One of the main effectors of IFNγ/TNFα-induced muscle loss is the iNOS/NO pathway (Di Marco *et al*, 2005, 2012; Hall *et al*, 2011). Here, not only do we confirm the importance of the iNOS/NO pathway in this process *in vivo*, but we also show that pY705-STAT3 is required for IFNγ/TNFα-mediated expression of iNOS. In response to IFNγ/TNFα, activated STAT3 associates with NF-κB in the cytoplasm and then translocates to the nucleus, where it is rapidly recruited to the *iNos* promoter. Taken together, this work demonstrates that different pro-cachectic inducers (i.e. IL-6 and IFNγ/TNFα) activate STAT3 signaling independently and that this signaling can collaborate with

the NF-κB pathway to induce cachexia through activation of target genes, like iNOS (Fig 7E). This suggests that although there is a diversity of inflammatory factors that can induce cachexia, there may be an integrated network of downstream effectors, such as STAT3-NFκB.

The importance of IL-6 and the mechanisms through which it promotes muscle wasting have been well established (Bonetto *et al*, 2011, 2012). However, evidence from clinical trials has clearly indicated that blocking IL-6 cannot completely stop or reverse muscle wasting (Bayliss *et al*, 2011). This may be due to other cytokines that are known to activate similar pathways as IL-6. Indeed, in this study, we found that treatment of muscle fibers with IFNγ/TNFα activates pY705-STAT3 resulting in muscle atrophy. Interestingly, we demonstrated, for the first time, that the activation of pY-STAT3 by IFNγ/TNFα during muscle wasting could occur independently of IL-6. Studies have also demonstrated that other pro-cachectic factors, such as transforming growth factor beta 1 (TGF-β1), can induce muscle wasting by activating STAT3 (Guadagnin *et al*, 2015). Thus, it is likely that monotherapies targeting any particular

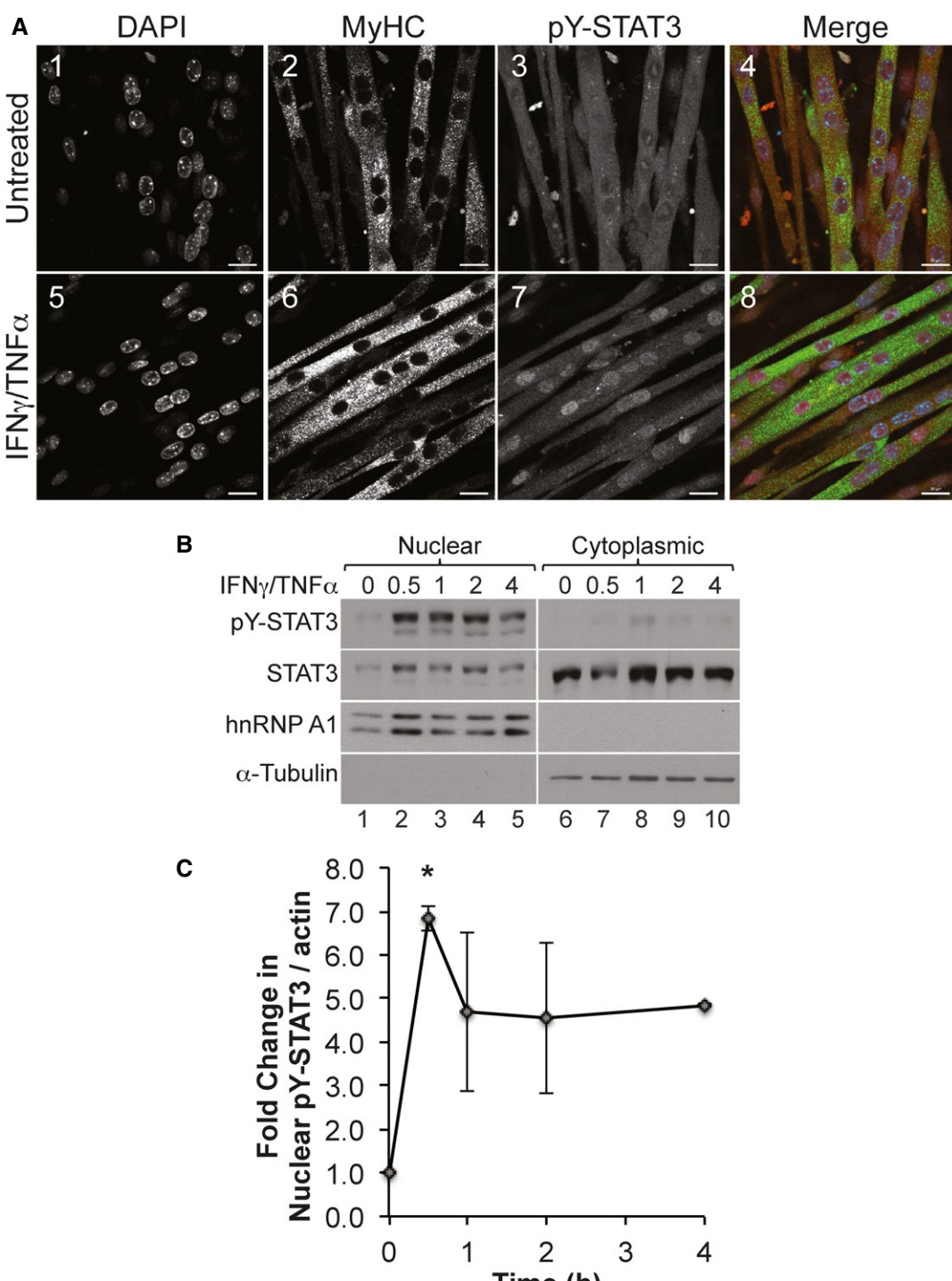

**Figure 6.   IFNγ/TNFα treatment induces a rapid translocation of pY-STAT3 to the nucleus.**

A   Confocal microscopy images of C2C12 myotubes treated with or without IFNγ/TNFα for 30 min and stained for pY-STAT3 (red), myosin heavy chain (MyHC; green), and counterstained with DAPI (blue). Scale bar = 20 μm. Images shown are representative of *n* = 3 experiments.

B   Nuclear and cytoplasmic fractions prepared from C2C12 myotubes treated with IFNγ/TNFα for the indicated amount of time were used for Western blot analysis with antibodies against pY-STAT3, total STAT3, p65, hnRNP A1 (nuclear marker), α-tubulin (cytoplasmic marker), and β-actin (loading control, see source data). Images shown are representative of *n* = 3 experiments.

C   Densitometric quantification of nuclear pY-STAT3 to β-actin in panel (B). Data represented as mean ± SEM (*n* = 3) with *P*-value = 0.0233 by one-way ANOVA with Dunnett's *post hoc* test.

Source data are available online for this figure.

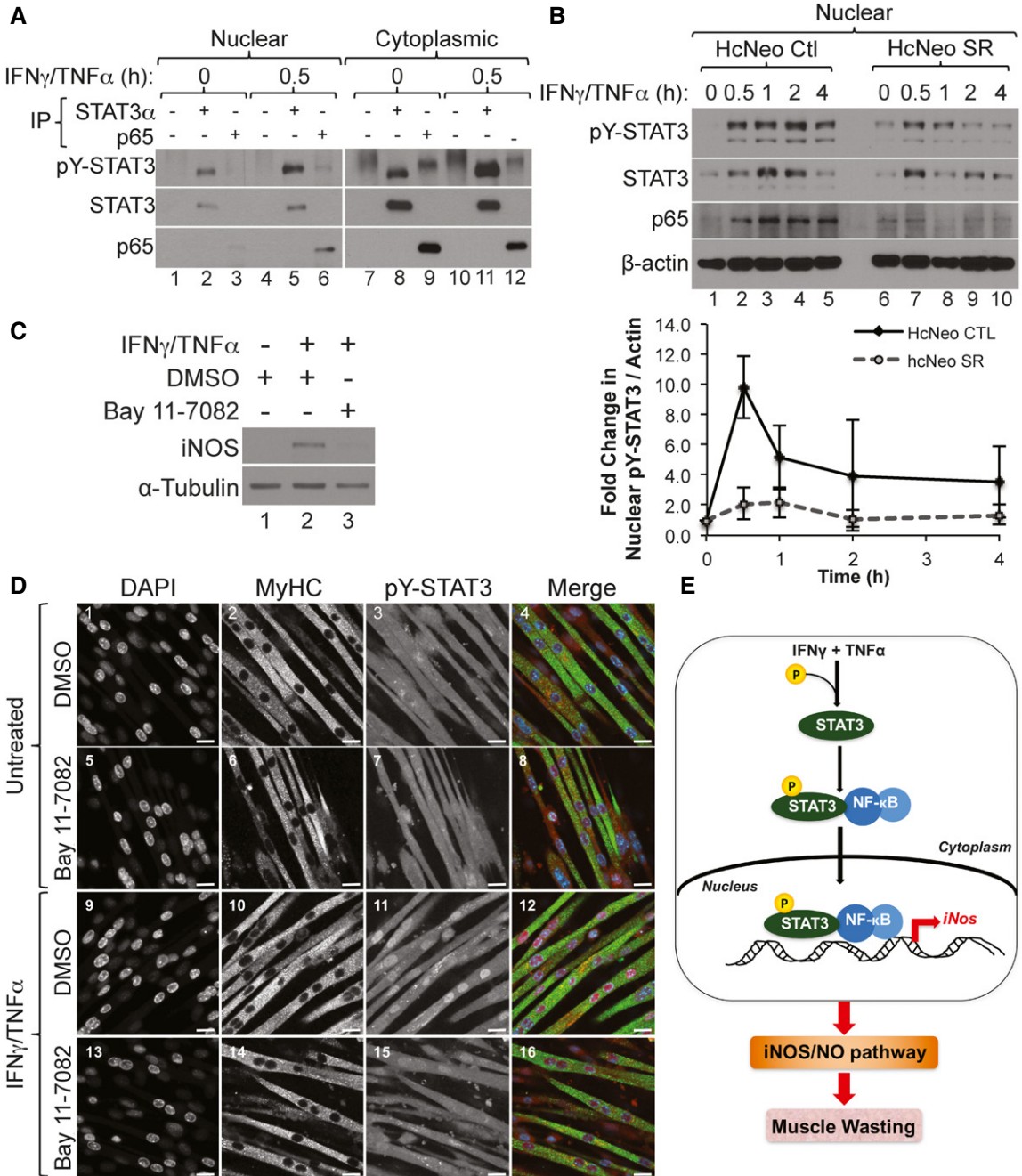

**Figure 7.  Active NF-κB pathway is required for the rapid translocation of pY-STAT3 to the nucleus during IFNγ/TNFα-induced muscle wasting.**

A   Nuclear and cytoplasmic fractions prepared from C2C12 myotubes treated with IFNγ/TNFα followed by immunoprecipitation (IP) with either total STAT3, p65, or an IgG control antibody. Western blot analysis was performed using antibodies against total STAT3, pY-STAT3, and p65. The blot shown is a representative of $n = 3$.

B   (Upper panel) Nuclear and cytoplasmic fractions were prepared from hcNeo control (Ctl) and hcNeo Super Repressor (SR) cells treated with IFNγ/TNFα and used for Western blot analysis with antibodies against pY-STAT3, total STAT3, p65, and β-actin. The blot shown is a representatives of $n = 2$. (Lower panel) Densitometric quantification of nuclear pY-STAT3 relative to β-actin. Data are represented as mean ± SD ($n = 2$).

C   Total cell extract from C2C12 myotubes pre-treated with Bay 11-7082 or DMSO as a control then treated with IFNγ/TNFα for 24 h were used for Western blot analysis with antibodies against iNOS and α-tubulin. The blot shown is a representative of $n = 2$.

D   Confocal microscopy images of C2C12 myotubes pre-treated with Bay 11-7082 or DMSO as a control, then treated with or without IFNγ/TNFα and stained for pY-STAT3 (red), MyHC (green), and counterstained with DAPI (blue). Scale bar = 20 μm. Images are representative of $n = 1$ experiment.

E   Model depicting the how STAT3 promotes cytokine-induced muscle wasting. The cytokines IFNγ and TNFα act synergistically by activating STAT3 via phosphorylation on its Y705 residue. Following the degradation of IκBα (not shown here), pY-STAT3 translocates to the nucleus as a complex with NF-κB and upregulates the expression of iNOS, leading to the activation of the iNOS/NO pathway, which in turn promotes muscle wasting.

Source data are available online for this figure.

factor, such as anti-IL-6 antibodies, would be unable to ablate STAT3 activation in cases where multiple cytokines are driving its activation. However, treatments that directly targeted STAT3 activation may be able to impair the transduction of pro-cachectic signaling from multiple cytokines simultaneously, potentially providing a much more robust and effective therapy.

While our work and others establishes STAT3 as a key mediator of muscle wasting, the mechanisms by which STAT3 induces atrophy remain elusive. Impairing STAT3 function has been shown to correlate with reduced protein proteolysis and muscle loss (Bonetto *et al*, 2012; Silva *et al*, 2015). However, STAT3 is a transcription factor, known to promote the expression of many target genes, and it is unknown which of these genes contribute to IFNγ/TNFα-induced muscle wasting. Here, we identify iNOS as a key target gene of STAT3-mediated cachexia. We found that STAT3 binds directly to the promoter of *iNos* to induce its expression. We have previously shown that iNOS is a key mediator of cytokine driven muscle wasting. Here, we show that iNOS-knockout mice are resistant to IFNγ/TNFα-induced muscle atrophy, suggesting that iNOS is a key gene required for STAT3-mediated wasting. Further studies are needed to identify the other genes that also contribute to the many facets of STAT3-induced muscle wasting.

Induction of gene expression by transcription factors often requires collaboration with partner proteins. Previous studies have found that during cancer-induced inflammation, STAT3 collaborates with other transcription factors such as NF-κB, also a well-known inducer of the cachectic phenotype (Fan *et al*, 2013). We and others have previously shown that the expression of iNOS depends on NF-κB (Di Marco *et al*, 2005; Altamirano *et al*, 2012). This suggests that collaboration between STAT3 and NF-κB could be an important step in promoting cachexia. Indeed, these two factors are already known central players in promoting inflammation (Bollrath & Greten, 2009; Grivennikov & Karin, 2010) and demonstrated to interact in the nucleus under inflammatory conditions (Yu *et al*, 2002b; Hagihara *et al*, 2005; Bode *et al*, 2012). Our study shows that the localization of pY-STAT3 to the nucleus of IFNγ/TNFα-treated fibers is likely to dependent on the degradation of IκBα and subsequent release of NF-κB, indicating the importance of cytoplasmic STAT3-NF-κB complex formation in facilitating the translocation of STAT3 and NF-κB to the nucleus. The nuclear localization of the STAT3-NF-κB-IκBα complex was recently shown to be regulated by Rac-1 in starved cancer cells (Kim & Yoon, 2016). Furthermore, it is known that the nuclear retention of NF-κB is prolonged by its interaction with STAT3, which is central to maintaining NF-κB activity in tumors (Lee *et al*, 2009). These results, together with our observations described above, suggest that cytoplasmic STAT3-NF-κB complex formation and its translocation are key steps in promoting muscle atrophy. Further investigation into the genes targeted by STAT3-NF-κB complex is needed to identify the networks of downstream genes through which these two transcription factors promote muscle wasting.

In our studies, we demonstrate that pY-STAT3 modulates IFNγ/TNFα-induced muscle wasting in an IL-6- and iNOS-dependent manner. Indeed, we showed that both IL-6- and iNOS-knockout mice, despite the phosphorylation of STAT3, are resistant to muscle wasting (Figs 2 and 4). This would suggest that both IL-6 and iNOS collaborate downstream of STAT3 activation to modulate IFNγ/TNFα-induced muscle atrophy. Indeed, we and others have shown

that both iNOS and IL-6 are targets of STAT3 transcriptional activity (Fig 4, Lee *et al*, 2013). Therefore, our results support the idea that targeting factors such as STAT3 may be a therapeutic avenue to prevent muscle wasting due to its ability to activate multiple downstream procachectic factors concurrently. Furthermore, our data suggests that the STAT3-NF-κB complex and their downstream target genes could help design novel therapeutic strategies to combat this deadly syndrome. Indeed, the fact that iNOS KO mice are protected from cytokine-induced muscle wasting is a clear indication that the iNOS/NO pathway should be considered as a potential target for such a strategy. Ultimately, the targeting of these integrated downstream effectors of the cachectic phenotype may prove more effective than mono-therapies against pro-cachectic inducers for the treatment of cachexia.

# Materials and Methods

### Cell culture and plasmids

C2C12 cells were obtained from American Type Culture Collection (VA, USA) and grown in Dulbecco's Modified Eagle Medium (DMEM, Invitrogen) containing high glucose, L-glutamine, sodium pyruvate, and supplemented with 20% fetal bovine serum (Sigma-Aldrich) and 1% penicillin/streptomycin (Sigma-Aldrich). Cells were routinely tested for mycoplasma. Cells were plated on tissue culture plates (Corning) coated with 0.1% gelatin (Sigma-Aldrich) for differentiation. To induce myotube formation, cells were grown to 100% confluency and switched to DMEM containing 2% horse serum (Gibco) and 1% penicillin/streptomycin for up to 4 days. Phase contrast pictures were taken with a 10× objective lens on an inverted Zeiss Axiovert 25 and a Sony Cyber-shot DSC-S75 Digital still camera.

A mixture of IFNγ (100 U/ml) (R&D Systems) and TNFα (20 ng/ml) (R&D Systems or BioBasic, Inc.) was used for the treatment of cells for various periods of time. S3I-201 and Jak Inhibitor 1 (EMD Millipore) were used to pre-treat cells as indicated. Short-hairpin RNA (shRNA) that specifically targets STAT3 or a scramble hairpin was purchased from Sigma. The pMXs-STAT3-C was a gift from Shinya Yamanaka (Addgene plasmid #13373) (Davis *et al*, 2006). STAT3 Y705F Flag pRC/CMV was a gift from Jim Darnell (Addgene plasmid #8709) (Wen & Darnell, 1997). C2C12 transfected with plasmid using jetPRIME™ DNA and siRNA transfection reagent (Polyplus Transfection) according to manufacturer's protocol.

### Study approval

The experiments using animal studies were approved by the McGill University Faculty of Medicine Animal Care Committee and comply with guidelines set by the Canadian Council of Animal Care.

### Mice

C56BL/6J wild-type or interleukin-6 knockout (IL-6 KO; B6.129S2-*Il6*^tm1Kopf/J, #002650) or inducible nitric oxide synthase KO mice (iNOS KO; B6.129P2-*Nos2*^tm1Lau/J, #002609, Jackson Laboratories) were housed in a room with a 12-h light–12-h dark cycle. All mice were housed in a sterile cage with corn-cob bedding and had free

access to water and rodent chow (#2920, Envigo). Testing for rodent-related pathogens was routinely performed by McGill University's Comparative Medicine and Animal Resources Center.

Male mice (21–25 g) aged 7–10 weeks were size-matched and randomly assigned to treatment with either saline (Sigma) or with 7,500 units of IFNγ (R&D Systems) and 3 μg of TNFα (R&D Systems). Mice were injected with a 30G needle every day for 5 days as described (Di Marco *et al*, 2012). On the sixth day, mice were sacrificed by $CO_2$ inhalation and the gastrocnemius muscle was dissected and frozen on dry ice when used for Western blot analysis. Muscle used for cross-sectional area analysis was frozen in isopentane (Sigma) chilled in liquid nitrogen. Each muscle was sectioned at 8 or 10 μm thickness and stained with hematoxylin and eosin. Wide-field images were taken with a 20× objective lens on an inverted Zeiss Axioskop microscope with an Axiocam MRc color camera in the McGill University Life Sciences Complex Advanced BioImaging Facility. The cross-sectional area (CSA) of muscle was analyzed blindly. Each animal was considered one experimental unit.

### Western blot analysis

Whole cell lysates were prepared by lysis in buffer containing 50 mM HEPES (pH 7.0), 150 mM NaCl, 10% glycerol, 1% Triton X-100, 10 mM sodium pyrophosphate, 100 mM NaF, 1 mM EGTA, 1.5 mM $MgCl_2$, 0.1 mM sodium orthovanadate, and complete EDTA-free protease inhibitors (Roche Applied Science). Primary antibodies used were phospho-Tyr$^{705}$-STAT3 (1:2,000; #9145, Cell Signaling), total STAT3 (1:1,500; #9132 or 9139, Cell Signaling), phospho-Ser$^{727}$-STAT3 (1:1,000; #9134, Cell Signaling), STAT3α (1:1,000; C-20, Santa Cruz), α-tubulin (1:2,000; clone 6G7 deposited by Halfter WM to Developmental Studies Hybridoma Bank), iNOS (1:3,000; Clone 6, BD Transduction Laboratories), hnRNP A1 (1:1,000; ab5832, Abcam), p65 (1:1,000; 06-418, EMD Millipore), and 3A2 [anti-HuR, 1:10,000 (Gallouzi *et al*, 2000)].

### Immunofluorescence

Cells were plated in 6-well plates coated with 0.1% gelatin and induced for differentiation. After treatment, cells were fixed in 3% paraformaldehyde for 30 min, permeabilized with 0.1% Triton X-100 in PBS, and then incubated with myoglobin (1:250; ab77232, Abcam) and myosin heavy chain (1:1,000; clone MF-20 deposited by Fischman DA to Developmental Studies Hybridoma Bank). Images were taken with a 40× objective lens on an inverted Zeiss Observer.Z1 microscope with an Axiocam MRm camera. The fiber widths were measured using the Axiovision software (release 4.8.2 SP2). Only fibers with at least three nuclei were measured in three places to obtain an average width for each fiber. At least three fields per condition for each experiment were measured. Measurements were analyzed in Microsoft Excel. For localization studies, cells were plated into an 8-well μ-slide (ibidi) coated with Matrigel (Corning) diluted to 1 mg/ml. Differentiated C2C12 myotubes were treated at various times and stained according to manufacturer's protocol. Images were taken with a 63× objective lens on an inverted Zeiss confocal laser scanning microscope 800 in the McGill University Life Sciences Complex Advanced BioImaging Facility.

### ELISA

The supernatant of IFNγ/TNFα-treated C2C12 myotubes was assayed for IL-6 production using the Mouse IL-6 ELISA Ready-SET-Go!$^{®}$ Kit (eBioscience, Inc.) according to the manufacturer's protocol.

### Quantitative PCR (qPCR)

One microgram of total RNA was reverse transcribed with the M-MuLV Reverse Transcriptase (New England Biolabs) according to the manufacturer's protocol. Each sample was diluted 1/20 and used to detect the mRNA levels of iNOS and RPL32. Expression of iNOS mRNA was normalized to RPL32 as a reference. The relative expression level was calculated using the $2^{-\Delta\Delta Ct}$ method, where $\Delta\Delta Ct$ is the difference in $C$t values between the target and reference genes. Primers for detection of mRNA are as follows: iNOS Forward: 5′-GTG CGC ATG GCT CGG GAT GT-3′, iNOS Reverse: 5′-GGC TGT CAG AGC CTC GTG GC-3′, RPL32 Forward: 5′-TTC TTC CTC GGC GCT GCC TAC GA-3′, and RPL32 Reverse: 5′-AAC CTT CTC CGC ACC CTG TTG TCA-3′.

### Subcellular fractionation

Pellets of C2C12 myotubes were immediately resuspended in EBKL buffer and incubated on ice for 15 min without agitation. The cellular membrane was lysed using a glass dounce with the tight pestle. The lysate was centrifuged, and the supernatant (cytoplasmic fraction) was removed. The pellet (nuclear fraction) was washed three times then lysed in nuclear lysis buffer. The cytoplasmic fraction was centrifuged again at 10,000 *g* to remove nuclear contamination.

### Co-immunoprecipitation

Fresh pellets of C2C12 myotubes were collected and immediately fractionated. 50 μl of protein A Sepharose beads pre-incubated with primary antibody for 4 h and washed three times before incubation with lysates overnight at 4°C. The following day, each IP was washed with once with low salt buffer and twice with medium salt buffer before adding equal volume of 2× Laemmli dye. Each sample was then vortexed and boiled. Samples were analyzed by Western blot analysis.

### Chromatin immunoprecipitation

One 10-cm dish of C2C12 myotubes was fixed in 1% formaldehyde and washed with PBS. Fixed cells were scraped in PBS, and cell pellets were frozen at −80°C. The Magna ChIP™ A/G One-Day Chromatin Immunoprecipitation Kit (Millipore) was used to perform ChIP according to manufacturer's protocol. Briefly, cell pellets were resuspended in 1 ml of cell lysis buffer and sonicated with a Branson Sonifier 450 attached to a cup horn for 30 s on/30 s off for 8 min total. 5% of each chromatin preparation was incubated with 2 μg of primary antibody against STAT3 or p65 or 1 μg of primary antibody against RNA polymerase II and 20 μl of protein A/G magnetic beads overnight. Then, beads were washed once with low

salt buffer, high salt buffer, LiCl buffer, and TE buffer. Protein–DNA complexes were eluted and reverse cross-linked with elution buffer containing proteinase K by rotating at 62°C for 2 h. DNA was isolated from the supernatant using the spin column provided and eluted in 50 μl of distilled $H_2O$. Using qPCR, 2 μl of ChIP DNA was amplified using the SsoFast EvaGreen Supermix (Bio-Rad). Raw $C$t values were analyzed using the fold enrichment method relative to IgG. The *iNOS/Nos2* promoter was detected using specific primer pair for the region containing putative STAT3 binding site (forward: 5′-CCAGAACAAAATCCCTCAGC-3′, reverse: 5′-CTCATGCAAGGC-CATCTCTT-3′) or the TATA box (forward: 5′-GAGCTAACTTGCA-CACCCAAC-3′, reverse: 5′-GCAGCAGCCATCAGGTATTT-3′).

Primary antibodies used for ChIP were against STAT3α (C-20X; Santa Cruz), p65 (ab7970, abcam), normal rabbit IgG (12-370, Millipore), RNA polymerase II (clone CTD4H8, #05-623, Millipore), and normal mouse IgG (12-371, Millipore).

### Detection of nitric oxide release

The detection of NO in the media was performed using Griess reagent as previously described. The O.D. was measured on a spectrophotometer at 543 nm (Di Marco *et al*, 2005).

### Statistics

Data are represented as mean ± SEM for $n = 3$ experiments unless otherwise stated. Quantification of band intensities was performed using ImageJ software (National Institute of Health). Statistical significance was evaluated using analysis of variance (ANOVA) or two-tailed, unpaired Student's $t$-test depending on the number of groups or factors to be analyzed. Data were considered statistically different with the following: * if $P < 0.05$, ** if $P < 0.01$, and *** if $P < 0.001$.

**Expanded View** for this article is available online.

### Acknowledgements

We are grateful to Erzsebet Nagy Kovacs for assistance with the animal studies. We thank Mr. Xian J. Lian for his technical help with immunoprecipitation experiments. This work is funded by a Prostate Cancer Canada Discovery Grant: D2014-14, a CIHR operating grant (MOP-142399), and a Qatar National Research Fund (QNRF) (NPRP8-457-3-101) to I.E.G. J.F.M. was supported by the CIHR/FRSQ training grant in cancer research FRN53888 of the McGill Integrated Cancer Research Training Program. D.T.H. was funded by a scholarship received from the Canadian Institute of Health Research (CIHR) funded Chemical Biology Program at McGill University.

### Author contributions

JFM, SDM, and I-EG contributed to the study conception and design. JFM, BJS, DTH, SDM, and I-EG contributed to the development of methodology. JFM, BJS, DTH, A-MKT, SDM, and I-EG were involved in the acquisition of data (provided animals, acquired and managed patients, provided facilities, etc.). JFM, SDM, and I-EG performed the analysis and interpretation of data (e.g. statistical analysis, bioStatistics, computational analysis). JFM, DTH, SDM, and I-EG wrote, reviewed, and/or revised the manuscripts. SDM and I-EG performed administrative, technical, or material support (i.e. reporting or organizing data, constructing databases). I-EG involved the study supervision.

### The paper explained

#### Problem

Cancer-induced muscle wasting, also known as cachexia, is a deadly syndrome characterized by progressive and involuntary muscle loss. In addition to its impact on patient quality of life, this syndrome is the direct cause of ~20% of all deaths resulting from various cancers. These patients frequently succumb to death due to collapse of respiratory muscles and heart function. As there is no effective cure for this syndrome, cancer cachexia has been (and still is) viewed as an end-of-life condition in patients with advanced and incurable malignancies and is managed primarily through palliative care approaches. Although pro-inflammatory cytokines, such as TNFα, IFNγ, and IL-6, are considered to be among the main drivers of cancer cachexia, none of the numerous therapeutic strategies that were developed to target these factors have shown any clear success.

#### Results

Our work provides an explanation for the failure of these monotherapy and identifies a cross talk between signaling pathways downstream of TNFα and IFNγ. We show that the STAT3 transcription factor, a well-known effector of IL-6-mediated muscle wasting, can also be activated by both TNFα and IFNγ independently of IL-6. In addition, in response to these two cytokine, STAT3 forms a complex with NF-κB, another transcription factor, which is mainly activated by TNFα. Upon formation, the STAT3-NF-κB complex translocates to the nucleus where it promotes the expression of pro-cachectic genes such as *iNos*.

#### Impact

Our work shows that various promoters of muscle wasting are able to activate common downstream signaling pathways. These observations shed light on some of the complex cross talk signaling events that underlie cachexia and highlight the multifactorial nature of this syndrome. In doing so, they suggest one of the potential reasons behind the failure of monotherapies to reverse muscle wasting is the integration of multiple signals into common downstream effectors. This opens the door for the consideration of downstream effectors, such as the NF-κB-STAT3 complex, as targets for the design of more effective therapeutic strategies to combat this deadly syndrome.

### Conflict of interest

The authors declare that they have no conflict of interest.

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
