## [Review Process File · EMBO Molecular Medicine]

STAT3 promotes IFN γ /TNF α -induced muscle wasting in an NF- κ B-dependent and IL-6-independent manner

Jennifer F. Ma, Brenda J. Sanchez, Derek T. Hall, Anne-Marie K. Tremblay, Sergio Di Marco, and Imed-Eddine Gallouzi

Corresponding author: Imed Gallouzi, McGill University

Review timeline:

Submission date:	10 September 2016
Editorial Decision:	11 October 2016
Revision received:	31 December 2016
Editorial Decision:	26 January 2017
Revision received:	04 February 2017
Accepted:	07 February 2017

Transaction Report:

Editor: Roberto Buccione

1st Editorial Decision

11 October 2016

Thank you for the submission of your manuscript to EMBO Molecular Medicine. We have now heard back from two Reviewers whom we asked to evaluate your manuscript.

We are sorry that it has taken so long to get back to you on your manuscript. In fact, we experienced unusual difficulties in securing three willing and appropriate reviewers and in obtaining their evaluations in a timely manner. I am now proceeding based on the two available evaluations to avoid further delays.

As you will see, the reviewers are both generally positive on your work, while expressing relevant concerns. Reviewer 1 is especially focused on the need to provide further experimental support and/or clarify aspects of the current dataset to validate the mechanistic conclusions. Reviewer 2 instead lists a number of items of concern regarding biological aspects. I should add that beyond these nicely complementary takes, they both converge on a number of specific points including on the effects of IFN γ alone and the role of NF κ B in STAT3 translocation to the nucleus. Reviewer 2 would also like to see further translational development.

After internal discussion and Reviewer cross commenting, it was agreed that although we will not be asking for additional experimentation to test inhibitors in a cachexia model (reviewer #2 point 6), the other concerns raised by the reviewers, mostly impinging on mechanistic and biological issues need to be carefully and fully addressed.

In conclusion, while publication of the paper cannot be considered at this stage, given the potential interest of your findings, we would be pleased to consider a revised submission, with the

understanding that the reviewers' concerns must be addressed with additional experimental data where appropriate as indicated above, and that acceptance of the manuscript will entail a second round of review.

I look forward to seeing a revised form of your manuscript in due time.

***** Reviewer's comments *****

Referee #1 (Comments on Novelty/Model System):

Cachexia is an important consequence of cancer, and may well be one of the main reasons for patient death. So far there is no known way to reverse or alleviate cancer cachexia. Thus, the finding that multiple pathways may converge in activating STAT3 and NF- κ B, leading to formation of a complex that drives iNOS transcription, is an important finding, especially coupled with the demonstration that blocking STAT3 phosphorylation prevents cachexia.

Referee #1 (Remarks):

This is a clear demonstration that IFN γ /TNF α can promote cachexia independently of IL-6 via activation of STAT3 and NF- κ B. But I have problems in understanding the mechanism, see below.

1. Experiments in Figs 1-3 are straightforward, save for one question: Can IFN γ alone induce cachexia? Would IFN γ plus any other NF- κ B inducer induce cachexia? This is not clear from the text, nor a clear reference is given to work that answers these questions.
2. Problems start with panel 4D. The STAT3 siRNA does not reduce STAT3 much, and there is no quantitation. With such a moderate reduction, iNOS should be induced, as it would happen in STAT3 heterozygotes. But a simple way to overcome these problems is to delete panel 4D. However:
3. in panel 4E, expression of STAT3-C alone does not induce iNOS, which is OK, but stimulation with TNF α does not induce iNOS either. Why? Shouldn't we expect STAT3-C to be sufficient, without additional IFN γ stimulation?
4. panels H-J in figure 4 are mislabeled: they should be labeled F-H.
5. Figure 5 needs an additional experiment: ChIP with anti-p65, and is possible ChIP-reChIP with anti-p65 and anti-STAT3.
6. In Fig 6B, why is beta-actin enriched in the nucleus relative to cytoplasm?
7. panel 7A is unreadable to me, save that anti-p65 brings down p65, and anti-STAT3 brings down STAT3. The status of pY-STAT3 does not make sense to me, also because some of the bands are not at the expected mw.

Finally, I have a basic question: why does STAT3, when phosphorylated, need NF- κ B to go to the nucleus? STAT3 phosphorylation and dimer formation should be more than sufficient.

Referee #2 (Remarks):

The present manuscript by Ma et al investigates the role of STAT3 and NF- κ B in skeletal muscle atrophy. They provide evidence that stimulation with IFN γ and TNF α promotes STAT3 phosphorylation on Tyrosine 705 in an IL-6-independent manner, then pSTAT3 binds NF- κ B and it translocates to the nucleus to regulate gene transcription. The authors have previously shown that iNOS is a relevant gene for skeletal muscle atrophy. Here they provide evidence that iNOS gene is a direct target of STAT3 in this context, and is required for the induction of muscle atrophy. This is an

interesting study that provides useful insights for the design of future therapeutic approaches for muscle wasting. The manuscript is well written and the data clearly presented. However, additional experiments and inclusion of relevant controls are required in order to fully support the authors' interpretation.

Main points:

1- In order to demonstrate a synergistic effect of IFN γ and TNF α , treatment with single factors, iFN γ alone and TNF α alone, should be performed in parallel, or it is difficult to ascribe the downstream events to synergistic interaction of the two cytokines.

2- In addition to reduction in myotube size, a striking phenotype observed in the results upon treatment with IFN γ and TNF α is the loss of myotubes, i.e. reduction in their numbers in the culture. Is the treatment inducing cell death (as one would expect with TNF α for example) and if so, how does this affect the remaining cells in the culture? Experiments assessing cell death in these conditions should be performed.

3- In Figure 2 the comparison between endogenous and recombinant IL-6 might suffer a bias associated with different stability and/or posttranslational modification of IL-6. A discussion of this aspect should be included in the text. The inclusion of experiments with the IL-6 $^{-/-}$ mouse are stronger, however the authors should show, in addition to STAT3 phosphorylation, what happens to the muscles upon treatment with IFN γ /TNF α , i.e. do they undergo atrophy? Assessment of myofiber cross-sectional area and muscle weight should be provided.

4- In Figure 4H-I the effects of IFN γ /TNF α on iNOS $^{-/-}$ mouse should be assessed by measuring myofibers cross-sectional area, not only muscle weight.

5- In Figure 5D-E, are the differences statistically significant? It is not indicated in the text or figure legend. In addition, as there are also NF- κ B binding sites on the iNOS promoter region, does NF- κ B bind the iNOS promoter region and cooperate with pSTAT3 for its transcriptional activation, as the authors indicate in the model in Figure 7E? If not, then what is the role of NF- κ B in this context, mainly translocating pSTAT3 into the nucleus, or does it have an effect on transcriptional regulation of genes associated with muscle atrophy? A more in depth evaluation of the role of NF- κ B in this context and assessment of genes involved in skeletal muscle atrophy would strengthen the findings.

6- Validating the findings by using specific inhibitors in a mouse model of atrophy would significantly increase the impact of the current work.

Specific Points

1- Figure 6A: the nuclear localization of pSTAT3 is not convincing. The quality of the images should be improved.

2- Immunoprecipitation in Figure 7A: it is unclear why the authors with p65 IP detect STAT3 in the nucleus, but not the other way around (IP of STAT3 does not detect p65)? Are the antibodies specific?

3- A statistics section should be included in the Methods.

1st Revision - authors' response

31 December 2016

We would like to thank the two reviewers for their thorough evaluation of our manuscript. We are grateful that both reviewers agreed that our findings are interesting and that the study addresses an important and medically relevant question.

Reviewer 1 "This is a clear demonstration that IFN γ /TNF α can promote cachexia independently of IL-6 via activation of STAT3 and NF- κ B".

Reviewer 2 "This is an interesting study that provides useful insights for the design of future

therapeutic approaches for muscle wasting”.

The two reviewers, however, raised several important issues that all have been addressed, as described below, both experimentally and by amending the text.

Referee #1

Cachexia is an important consequence of cancer, and may well be one of the main reasons for patient death. So far there is no known way to reverse or alleviate cancer cachexia.

Thus, the finding that multiple pathways may converge in activating STAT3 and NF- κ B, leading to formation of a complex that drives iNOS transcription, is an important finding, especially coupled with the demonstration that blocking STAT3 phosphorylation prevents cachexia.

Remarks:

This is a clear demonstration that IFN γ /TNF α can promote cachexia independently of IL-6 via activation of STAT3 and NF- κ B.

But I have problems in understanding the mechanism, see below.

- 1- Experiments in Figs 1-3 are straightforward, save for one question: Can IFN γ alone induce cachexia? Would IFN γ plus any other NF- κ B inducer induce cachexia? This is not clear from the text, nor a clear reference is given to work that answers these questions.

We thank the reviewer for this comments that we addressed experimentally. We treated C2C12 myotubes with TNF α or IFN γ separately (see new Fig. S1). Unlike both cytokines together, exposing C2C12 myotubes to either TNF α or IFN γ for 72h or 96h did not trigger muscle wasting. This result is consistent with previous observations (Guttridge et al., Science, 2000). Additionally, as suggested by the reviewer we included in the result section a statement describing these results as well as a reference to support the observation (see result section, page 6, lanes 6 and 7).

- 2- Problems start with panel 4D. The STAT3 siRNA does not reduce STAT3 much, and there is no quantitation. With such a moderate reduction, iNOS should be induced, as it would happen in STAT3 heterozygotes. But a simple way to overcome these problems is to delete panel 4D.

The reviewer is correct for raising this point, and for that we are grateful. We repeated this experiment two more times and a more representative western blot is now shown. We observed that indeed the expression level of iNOS is reduced proportionally to the STAT3 knockdown. Moreover, we now provide quantification that reflects this fact (see the new Figure 4D). A statement describing these results was included in the result section (see page 8, lanes 27-30).

However:

- 3- in panel 4E, expression of STAT3-C alone does not induce iNOS, which is OK, but stimulation with TNF α does not induce iNOS either. Why? Shouldn't we expect STAT3-C to be sufficient, without additional IFN γ stimulation?

We agree with the reviewer that in principle, TNF α alone should induce iNOS expression. However, since we began working, few years ago, on the role of cytokines in muscle wasting it has been difficult for us to observe a great induction of iNOS in C2C12 myotubes with TNF α alone (Di Marco et al., MCB 2005). Usually we observe only a small induction of iNOS in myotubes with TNF α alone. However, others and we have always observed that both TNF α and IFN γ synergistically induce a high level of iNOS in C2C12 myotubes as early 18h after exposure. To illustrate this point in the revised

version of the manuscript, we included a new experiment (Fig. S5). We show that in the presence of TNF α alone for 48h but not IFN γ , iNOS becomes detectable in C2C12 myotubes. However, and as we observed before (Di Marco et al., MCB 2005), only both TNF α and IFN γ together trigger a significant expression level of iNOS. Therefore, based on these and our previous observation, we strongly believe that in muscle fibers a collaboration between several factors/pathways is needed to promote iNOS expression. A statement describing these results is now included in this revised version of the paper (see result section, page 8, lanes 21-25).

Additionally, the data in this paper clearly show that the collaboration between NF- κ B and STAT3 is required but likely not sufficient to promote iNOS. This point was clarified in the discussion section.

- 4- panels H-J in figure 4 are mislabeled: they should be labeled F-H.

This was corrected. Figure 4 now includes picture and statistic of cross section area of the gastrocnemius of Wild-type and iNOS $-/-$ mice as Figures 4H and 4I (see result section, page 9, lanes 6-13).

- 5- Figure 5 needs an additional experiment: ChIP with anti-p65, and is possible ChIP-reChIP with anti-p65 and anti-STAT3.

We thank the reviewer for suggesting this additional experiment that we successfully performed. We observed that, similarly to STAT3, p65 associates with iNOS promoter. An additional panel (Fig. 5E) is now included as part of Figure 5 of the revised manuscript. However, despite our numerous attempts, we were not successful in obtaining clear and interpretable results with Chip-reChip experiments (see result section, page 9, lanes 23-29).

- 6- In Fig 6B, why is beta-actin enriched in the nucleus relative to cytoplasm?

This is a good point. While others and we consistently see b-actin in the nucleus (Dormoy-Raclet et al., MCB 2007), as far as we know the reason behind its high levels in the nucleus is not well-understood. However, numerous studies have reported that b - actin is implicated in many events that occur in the nucleus such as transcription, nucleocytoplasmic transport, and chromatin and nuclear structure [Bettinger et al, Nature Reviews Molecular Cell Biology 5, 410-415 (May 2004)]. A statement to this effect is now included in the result section (see result section, page 10, lanes 10-13).

- 7- panel 7A is unreadable to me, save that anti-p65 brings down p65, and anti-STAT3 brings down STAT3. The status of pY-STAT3 does not make sense to me, also because some of the bands are not at the expected mw.

We are grateful to the reviewer for raising this issue. We repeated the experiments and we now provide a much better blot demonstrating the immunoprecipitation of STAT3, p-STAT3 and p65 from nuclear fraction (new Figure 7A).

Finally, I have a basic question: why does STAT3, when phosphorylated, need NF- κ B to go to the nucleus? STAT3 phosphorylation and dimer formation should be more than sufficient.

This is a great question. Of course, based on what we know regarding STAT3 function, we agree with the reviewer "that STAT3 phosphorylation and dimer formation should be more than sufficient" to translocate to the nucleus and induce gene expression. However, as mentioned in the introduction and in the discussion sections of the paper, there are strong evidence that STAT3 and NF- κ B collaborate in macrophages in response to either TNF α or IFN γ . Both NF- κ B and STAT3 are transcription factors and both translocate to the nucleus to trigger gene expression. Our data however, strongly suggest that under conditions where inflammation is driving muscle loss by activating these two pathways simultaneously, it makes sense that both collaborate to go to the nucleus to trigger the expression of common pro-cachectic genes. Of course, our finding just open the door to consider this possibility and much more needs to be done to

understand how these two pathways converge to promote muscle wasting in vivo.

Referee #2

The present manuscript by Ma et al investigates the role of STAT3 and NF-kb in skeletal muscle atrophy. They provide evidence that stimulation with IFN γ and TNF α promotes STAT3 phosphorylation on Tyrosine 705 in an IL-6-independent manner, then pSTAT3 binds NF-kb and it translocates to the nucleus to regulate gene transcription. The authors have previously shown that iNOS is a relevant gene for skeletal muscle atrophy. Here they provide evidence that iNOS gene is a direct target of STAT3 in this context, and is required for the induction of muscle atrophy. This is an interesting study that provides useful insights for the design of future therapeutic approaches for muscle wasting. The manuscript is well written and the data clearly presented. However, additional experiments and inclusion of relevant controls are required in order to fully support the authors' interpretation.

Main points:

- 1- In order to demonstrate a synergistic effect of IFN γ and TNF α , treatment with single factors, IFN γ alone and TNF α alone, should be performed in parallel, or it is difficult to ascribe the downstream events to synergistic interaction of the two cytokines.

We thank the reviewer for this comments that was also raised by reviewer 1. As described above, we have addressed this issue both experimentally and by amending the text (see new Fig. S1). We treated C2C12 myotubes with TNF α or IFN γ separately. Unlike both cytokines together, exposing C2C12 myotubes to either TNF α or IFN γ for 72h or 96h did not trigger muscle wasting. This result is consistent with previous observations (Guttridge et al., Science, 2000). Additionally, as suggested by the reviewer we included in the result section a statement describing these results as well as added a reference to support the observation (see result section, page 6, lanes 6 and 7).

- 2- In addition to reduction in myotube size, a striking phenotype observed in the results upon treatment with IFN γ and TNF α is the loss of myotubes, i.e. reduction in their numbers in the culture. Is the treatment inducing cell death (as one would expect with TNF α for example) and if so, how does this affect the remaining cells in the culture? Experiments assessing cell death in these conditions should be performed.

This is a great point. As requested we assessed apoptosis in myotubes exposed to TNF α and IFN γ . We followed caspase 3 cleavage and observed that TNF α and IFN γ -induced loss of myotubes is not due to the activation of the caspase-dependent apoptotic pathway (see the new Fig. S2 and result section, page 6, lanes 9-12).

- 3- In Figure 2 the comparison between endogenous and recombinant IL-6 might suffer a bias associated with different stability and/or posttranslational modification of IL-6. (a) A discussion of this aspect should be included in the text. (b) The inclusion of experiments with the IL-6 $^{-/-}$ mouse are stronger, however the authors should show, in addition to STAT3 phosphorylation, what happens to the muscles upon treatment with IFN γ and TNF α , i.e. do they undergo atrophy? Assessment of myofiber cross-sectional area and muscle weight should be provided.

(a) We thank the reviewer for raising this point. Our experiments clearly show that only high levels of the recombinant IL-6 used (100ng/ml) promotes the phosphorylation of STAT3 in macrophages and C2C12 myotubes (see Fig. S4). Therefore, in our opinion the absence of STAT3 activation in myotubes by low levels of IL-6 treated is not due to difference in "stability and/or posttranslational modification of recombinant IL-6. A statement highlighting this fact was included in the result section (see result section, page 7, lanes 10-12).

(b) The experiments with IL-6 $^{-/-}$ mice was repeated as requested. We now show as part of Figure 2 of the revised manuscript that, in IL-6 $^{-/-}$ mice while TNF α and IFN γ

activate STAT3 they do not, as expected, trigger muscle loss. We provide myofiber cross-sectional area and muscle weight to support this conclusion (see new Fig 2 F-H panels). We included a statement in the result section to highlight these new data (see page 7, lanes 17-19)

- 4- In Figure 4H-I the effects of IFN γ /TNF α on iNOS $^{-/-}$ mouse should be assessed by measuring myofibers cross-sectional area, not only muscle weight.

We agree with the reviewer and we included, in the revised manuscript, cross-sectional area analysis. The new Figure 4G-I now show myofibers cross-sectional area of the gastrocnemius of WT and iNOS $^{-/-}$ treated or not with TNF α /IFN γ . As expected from previous observation (Di Marco et al., MCB 2005), these mice are protected against TNF α /IFN γ -induced wasting.

- 5- In Figure 5D-E, are the differences statistically significant? It is not indicated in the text or figure legend. In addition, as there are also NF-kB binding sites on the iNOS promoter region, does NF-kB bind the iNOS promoter region and cooperate with pSTAT3 for its transcriptional activation, as the authors indicate in the model in Figure 7E?

If not, then what is the role of NF-kB in this context, mainly translocating pSTAT3 into the nucleus, or does it have an effect on transcriptional regulation of genes associated with muscle atrophy? A more in depth evaluation of the role of NF-kB in this context and assessment of genes involved in skeletal muscle atrophy would strengthen the findings.

We thank the reviewer for pointing out this omission on our part, an issue that that was also raised by reviewer 1. We performed the Chip experiment using the anti-p65 antibody and showed that, similarly to p-STAT3, NF-kB associates with iNOS promoter (See new Fig. 5E). The figure legend now includes information regarding the statistic and significance.

- 6- Validating the findings by using specific inhibitors in a mouse model of atrophy would significantly increase the impact of the current work.

While we agree with the reviewer that validating the use of STAT3 and NF-kB inhibitors in vivo model with both TNF α and IFN γ will strengthen the paper, the optimization of such an approach will go beyond the time frame allocated for a normal round of revision. We hope that the reviewer agrees with us as well as with the editor who has indicated that this experiment is not needed for this study.

Specific Points

- 1- Figure 6A: the nuclear localization of pSTAT3 is not convincing. The quality of the images should be improved.

We are now providing a better picture that clearly show the nuclear translocation of p-STAT3 in response to TNF α /IFN γ treatment (see the new Fig. 6A).

- 2- Immunoprecipitation in Figure 7A: it is unclear why the authors with p65 IP detect STAT3 in the nucleus, but not the other way around (IP of STAT3 does not detect p65)? Are the antibodies specific?

Indeed, in spite our numerous attempts by trying few different anti-STAT3 antibodies we were not able to immunoprecipitate p65 with STAT3.

- 3- A statistics section should be included in the Methods.

We apologize for this omission and as requested a section describing statistic methods used for our analysis is now included in the materials and methods sections.

Thank you for the submission of your revised manuscript to EMBO Molecular Medicine. We have now received the enclosed reports from the Reviewers who were asked to re-assess it.

As you will see, the reviewers are now satisfied pending a few remaining concerns for you to deal with. I am prepared to evaluate your next, final version of your manuscript at the editorial level, provided the items are fully addressed. Please highlight the text amendments in the revised manuscript and a rebuttal.

Please also comply with the following editorial amendments::

- 1) Please upload separate files for each figure and provide the appendix as a pdf file.
- 2) While performing our pre-acceptance quality control and image screening routines, we noted vertical markings of unclear origin in Fig. 3F and Fig. 7B (much more prominent). Please provide improved images and explain these occurrences. In connection to this, please also consider point 4 below.
- 3) Every published paper now includes a 'Synopsis' to further enhance discoverability. Synopses are displayed on the journal webpage and are freely accessible to all readers. They include a short standfirst (to be written by the editor) as well as 2-5 one sentence bullet points that summarise the paper (to be written by the author). Please provide the short list of bullet points that summarise the key NEW findings. The bullet points should be designed to be complementary to the abstract - i.e. not repeat the same text. We encourage inclusion of key acronyms and quantitative information. Please use the passive voice. Please attach these in a separate file or send them by email, we will incorporate them accordingly.
- 4) We are now encouraging the publication of source data, particularly for electrophoretic gels and blots, with the aim of making primary data more accessible and transparent to the reader. Would you be willing to provide a PDF file per figure that contains the original, uncropped and unprocessed scans of all or at least the key gels used in the manuscript? The PDF files should be labeled with the appropriate figure/panel number, and should have molecular weight markers; further annotation may be useful but is not essential. The PDF files will be published online with the article as supplementary "Source Data" files. If you have any questions regarding this just contact me.

Please submit your revised manuscript within two weeks. I look forward to seeing a revised form of your manuscript as soon as possible.

***** Reviewer's comments *****

Referee #1 (Remarks):

The manuscript is much improved, and the authors have answered in a substantive way to the queries.

I just find a few details that can be improved by text changes, as detailed below:

The title of suppl Fig 1 is wrong: a "not" is missing. This suppl figure should go into the main text, as part of Fig 1.

The conclusion that low levels of IL-6 are not due to differences between natural and recombinant IL-6 is not fully substantiated. My suggestion is to say "However, pY-STAT3 was detected when C2C12 muscle fibers or macrophages (used as positive control) were treated with 100ng/mL of IL-6 (Fig. S4). Therefore, the failure of low doses of rIL-6 (20ng/ml or lower) to phosphorylate STAT3 in myotubes might be due to instability or inactivity of our rIL-6." This further motivates the experiment with IL-6 KO, which is the critical experiment.

The sentence "The effect of the inhibitor on iNOS expression is due to affects on STAT3 since an shRNA that reduced STAT3 expression by ~40% also decreased iNOS protein levels by >35% in muscle fibers treated with IFN γ /TNF α (Fig 4D)." is logically incorrect. Just say "Moreover, an shRNA that reduced STAT3 expression by ~40% also decreased iNOS protein levels by >35% in muscle fibers treated with IFN γ /TNF α (Fig 4D)."

In Fig. 6B, while tubulin defines that there is no contamination of the nuclear fraction, the presence of more actin in the nuclear fraction than in the cytoplasmic fraction is actually confusing (although not wholly unexpected, I agree with the authors). The tubulin and the hnRNP A1 markers are sufficient, I suggest to just delete the actin western blot.

Referee #2 (Remarks):

The authors have addressed most of my comments and as a result the manuscript is significantly strengthened. I only have one remaining comment:

In the revised manuscript the authors have now included, as requested, in addition to STAT3 phosphorylation, data on the effects of IFN/TNF in vivo treatment on IL-6 $^{-/-}$ skeletal muscle, and show that, while there is robust STAT3 phosphorylation and iNOS activation even in the absence of IL-6, the muscles are resistant to atrophy. This aspect should be discussed in the text, i.e. why do the authors think that, in spite of proper STAT3 phosphorylation and iNOS activation, this model lacking IL-6 is resistant to atrophy? Why is IFN/TNF treatment not sufficient in this context to induce atrophy?

2nd Revision - authors' response

04 February 2017

Point-by-point rebuttal to editor's/reviewers' comments

We would like to thank the editor and the two reviewers for their thorough review of our revised manuscript. We are thankful and grateful to the editor and reviewers for their support of the publication of our manuscript once we have addressed their minor comments.

As requested by the editor all the amendments are highlighted in yellow in the text.

Reviewers comments provided below are shown in bold, italic font while our responses are in normal font.

Editor's comments:

- 1) Please upload separate files for each figure. Provide the appendix as a pdf file.***

As requested, we have uploaded the figures as separate files and the appendix as a PDF file.

- 2) While performing our pre-acceptance quality control and image screening routines, we noted vertical markings of unclear origin in Fig. 3F and Fig. 7B (much more prominent). Please provide improved images and explain these occurrences. In connection to this, please also consider point 4 below.***

The vertical line that appears in Fig.3F is due to the apparatus that was used to develop the western blot films (as seen below as well as in the source files) at the time these experiments were performed. The vertical lines seen in Fig. 7B, on the other hand, surfaced during the scanning of the western blot film. We have rescanned these films and have provided, in the re-revised manuscript and in the source files, new scans/figure that no longer shows these markings.

New Figure 3F

- 3) *Every published paper now includes a 'Synopsis' to further enhance discoverability. Synopses are displayed on the journal webpage and are freely accessible to all readers. They include a short standfirst (to be written by the editor) as well as 2-5 one sentence bullet points that summarise the paper (to be written by the author). Please provide the short list of bullet points that summarise the key NEW findings. The bullet points should be designed to be complementary to the abstract - i.e. not repeat the same text. We encourage inclusion of key acronyms and quantitative information. Please use the passive voice. Please attach these in a separate file or send them by email, we will incorporate them accordingly.*

As requested, we have provided a synopsis for our manuscript.

- 4) *We are now encouraging the publication of source data, particularly for electrophoretic gels and blots, with the aim of making primary data more accessible and transparent to the reader. Would you be willing to provide a PDF file per figure that contains the original, uncropped and unprocessed scans of all or at least the key gels used in the manuscript? The PDF files should be labeled with the appropriate figure/panel number, and should have molecular weight markers; further annotation may be useful but is not essential. The PDF files will be published online with the article as supplementary "Source Data" files. If you have any questions regarding this just contact me.*

As requested, we have provided the source data for all Figures in the main text of the manuscript (containing our original uncropped and unprocessed scans).

***** Reviewer's comments *****

Referee #1 (Remarks):

- 1) *The title of suppl Fig 1 is wrong: a "not" is missing. This suppl figure should go into the main text, as part of Fig 1.*

As requested, we have moved suppl.figure 1 (in the revised manuscript) to Figure 1 of our revised manuscript (Figure 1C of the re-revised manuscript).

- 2) *The conclusion that low levels of IL-6 are not due to differences between natural and recombinant IL-6 is not fully substantiated. My suggestion is to say "However, pY-STAT3 was detected when C2C12 muscle fibers or macrophages (used as positive control) were treated*

with 100ng/mL of IL-6 (Fig. S4). Therefore, the failure of low doses of rIL-6 (20ng/ml or lower) to phosphorylate STAT3 in myotubes might be due to instability or inactivity of our rIL-6." This further motivates the experiment with Il-6 KO, which is the critical experiment.

We have modified our text (highlighted on page 7 of the re-revised manuscript) to include the suggestion proposed by the author.

3) The sentence "The effect of the inhibitor on iNOS expression is due to affects on STAT3 since an shRNA that reduced STAT3 expression by ~40% also decreased iNOS protein levels by >35% in muscle fibers treated with IFN γ /TNF α (Fig 4D)." is logically incorrect. Just say "Moreover, an shRNA that reduced STAT3 expression by ~40% also decreased iNOS protein levels by >35% in muscle fibers treated with IFN γ /TNF α (Fig 4D)."

We have modified our text (highlighted on page 8 of the re-revised manuscript) to include the suggestion proposed by the author.

3) In Fig. 6B, while tubulin defines that there is no contamination of the nuclear fraction, the presence of more actin in the nuclear fraction than in the cytoplasmic fraction is actually confusing (although not wholly unexpected, I agree with the authors). The tubulin and the hnRNP A1 markers are sufficient, I suggest to just delete the actin western blot.

The actin western blot, as suggested by the reviewer, was deleted from the re-revised manuscript. It was nonetheless kept in the source data since, as indicated in the legend for the figure, the pY-STAT3 levels were normalized to the actin levels. A statement to this effect is included in the manuscript (See Figure legend for 6B, page 34 of re-revised manuscript).

Referee #2 (Remarks):

The authors have addressed most of my comments and as a result the manuscript is significantly strengthened. I only have one remaining comment:

In the revised manuscript the authors have now included, as requested, in addition to STAT3 phosphorylation, data on the effects of IFN/TNF in vivo treatment on IL-6-/- skeletal muscle, and show that, while there is robust STAT3 phosphorylation and iNOS activation even in the absence of IL-6, the muscles are resistant to atrophy. This aspect should be discussed in the text, i.e. why do the authors think that, in spite of proper STAT3 phosphorylation and iNOS activation, this model lacking IL-6 is resistant to atrophy? Why is IFN/TNF treatment not sufficient in this context to induce atrophy?

Muscle wasting is known occur due to the activation of multiple factors that, collectively, activate pathways that result in muscle loss. The results presented in Figs. 2D-H as well as Figs. 4F-I provide evidence suggesting that this is the case. We demonstrate in these figures that muscle wasting was not observed in IL-6 and iNOS knockout mice despite the phosphorylation of STAT3. Although the premise of the manuscript is that IFN/TNF-induced STAT3-phosphorylation does occur independently of IL-6, we believe that NF-kB and STAT3 cooperate under these conditions to induce the expression of IL-6 and iNOS that, consequently, collaborate to activate additional downstream pathways required for muscle wasting. Indeed, we as well as others have demonstrated, to this effect, that IFN/TNF induces both iNOS and IL-6 production in C2C12 muscle fibers (Figs. 2,4 of the re-revised manuscript) and that both are regulated transcriptionally by STAT3 and NF-kB in a cooperative manner (Figs. 4, Di Marco et.al, 2005, Yu et.al, 2002, Yoon, et.al, 2012). We have included, in the discussion of the revised manuscript (page 14), an explanation to this effect.

Corresponding Author Name: Imed Gallouzi
Journal Submitted to: EMBO Molecular Medicine
Manuscript Number: 2016-07052